# CONDITIONAL OBJECT-CENTRIC LEARNING FROM VIDEO

**Thomas Kipf** *†**, Gamaleldin F. Elsayed*, Aravindh Mahendran*, Austin Stone*,
Sara Sabour, Georg Heigold, Rico Jonschkowski, Alexey Dosovitskiy & Klaus Greff**
Google Research

## ABSTRACT

Object-centric representations are a promising path toward more systematic generalization by providing flexible abstractions upon which compositional world models can be built. Recent work on simple 2D and 3D datasets has shown that models with object-centric inductive biases can learn to segment and represent meaningful objects from the statistical structure of the data alone without the need for any supervision. However, such fully-unsupervised methods still fail to scale to diverse realistic data, despite the use of increasingly complex inductive biases such as priors for the size of objects or the 3D geometry of the scene. In this paper, we instead take a weakly-supervised approach and focus on how 1) using the temporal dynamics of video data in the form of optical flow and 2) conditioning the model on simple object location cues can be used to enable segmenting and tracking objects in significantly more realistic synthetic data. We introduce a sequential extension to Slot Attention which we train to predict optical flow for realistic looking synthetic scenes and show that conditioning the initial state of this model on a small set of hints, such as center of mass of objects in the first frame, is sufficient to significantly improve instance segmentation. These benefits generalize beyond the training distribution to novel objects, novel backgrounds, and to longer video sequences. We also find that such initial-state-conditioning can be used during inference as a flexible interface to query the model for specific objects or parts of objects, which could pave the way for a range of weakly-supervised approaches and allow more effective interaction with trained models.

Project page: https://slot-attention-video.github.io/

## 1 INTRODUCTION

Humans understand the world in terms of separate *objects* (Kahneman et al., 1992; Spelke & Kinzler, 2007), which serve as compositional building blocks that can be processed independently and recombined. Such a compositional model of the world forms the foundation for high-level cognitive abilities such as language, causal reasoning, mathematics, planning, etc. and is crucial for generalizing in predictable and systematic ways. Object-centric representations have the potential to greatly improve sample efficiency, robustness, generalization to new tasks, and interpretability of machine learning algorithms (Greff et al., 2020). In this work, we focus on the aspect of modeling motion of objects from video, because of its synergistic relationship with object-centric representations: On the one hand, objects support learning an efficient dynamics model by factorizing the scene into approximately independent parts with only sparse interactions. Conversely, motion provides a powerful cue for which inputs should be grouped together, and is thus an important tool for learning about objects.

Unsupervised multi-object representation learning has recently made significant progress both on images (e.g. Burgess et al., 2019; Greff et al., 2019; Lin et al., 2020; Locatello et al., 2020) and on video (e.g. Veerapaneni et al., 2020; Weis et al., 2020; Jiang et al., 2020). By incorporating object-centric inductive biases, these methods learn to segment and represent objects from the statistical structure of the data alone without the need for supervision. Despite promising results these methods are currently limited by two important problems: Firstly, they are restricted to toy data like moving 2D sprites or very simple 3D scenes and generally fail at more realistic data with complex textures (Greff

---

*Equal contribution. †Correspondence to: tkipf@google.com

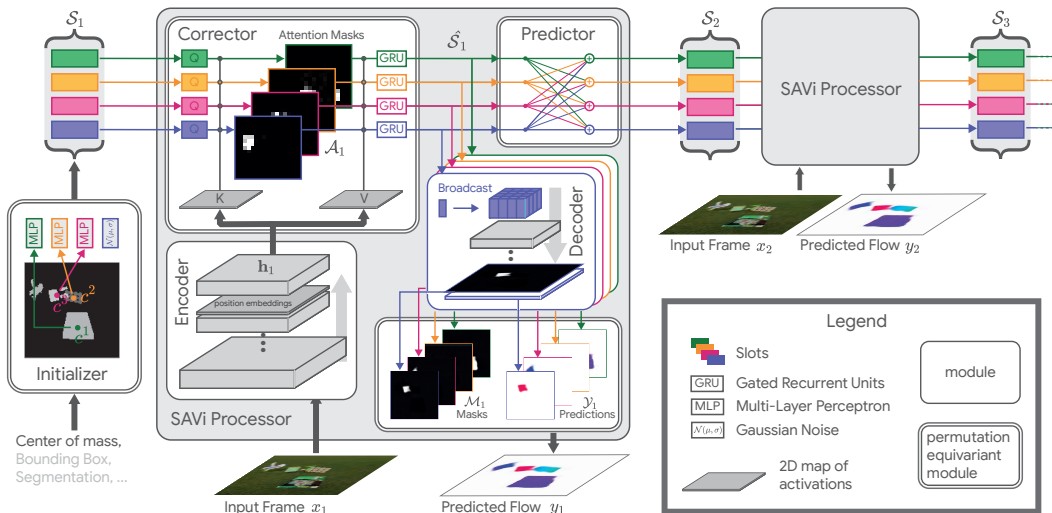

Figure 1: Slot Attention for Video (SAVi) architecture overview. The SAVi Processor is recurrently applied to a sequence of input video frames. SAVi maintains a set $\mathcal{S}_t = [\boldsymbol{s}_t^1, \ldots, \boldsymbol{s}_t^K]$ of $K$ latent slot representations at each time step $t$. Slots can be *conditionally* initialized based on cues such as center of mass coordinates of objects and subsequently learn to track and represent a particular object. SAVi is trained to predict optical flow or to reconstruct input frames.

et al., 2019; Harley et al., 2021; Karazija et al., 2021). And secondly, it is not entirely clear how to interface with these models both during training and inference. The notion of an object is ambiguous and task-dependent, and the segmentation learned by these models does not necessarily align with the tasks of interest. The model might, for example, over-segment a desired object into distinct parts, or alternatively fail to segment into the desired parts. Ideally, we would like to be able to provide the model with hints as to the desired level of granularity during training, and flexibly query the model during inference to request that the model detects and tracks a particular object or a part thereof.

In this paper we introduce a sequential extension of Slot Attention (Locatello et al., 2020) that we call *Slot Attention for Video* (SAVi) to tackle the problem of unsupervised / weakly-supervised multi-object segmentation and tracking in video data. We demonstrate that 1) using optical flow prediction as a self-supervised objective and 2) providing a small set of abstract hints such as the center of mass position for objects as conditional inputs in the first frame suffices to direct the decomposition process in complex video scenes without otherwise requiring any priors on the size of objects or on the information content of their representations. We show successful segmentation and tracking for synthetic video data with significantly higher realism and visual complexity than the datasets used in prior work on unsupervised object representation learning. These results are robust with respect to noise on both the optical flow signal and the conditioning, and that they generalize almost perfectly to longer sequences, novel objects, and novel backgrounds.

## 2   SLOT ATTENTION FOR VIDEO (SAVI)

We introduce a sequential extension of the Slot Attention (Locatello et al., 2020) architecture to video data, which we call SAVi. Inspired by predictor-corrector methods for integration of ordinary differential equations, SAVi performs two steps for each observed video frame: a prediction and a correction step. The correction step uses Slot Attention to update (or *correct*) the set of slot representations based on slot-normalized cross-attention with the inputs. The prediction step uses self-attention among the slots to allow for modeling of temporal dynamics and object interactions. The output of the predictor is then used to initialize the corrector at the next time step, thus allowing the model to consistently track objects over time. Importantly, both of these steps are permutation equivariant and thus are able to preserve slot symmetry. See Figure 1 for a schematic overview of the SAVi architecture.

**Encoder**   For each time-step $t \in \{1 \ldots T\}$ the corresponding video frame $\boldsymbol{x}_t$ is first passed through a small convolutional neural network (CNN) encoder (here, a stack of five convolutional layers with

ReLUs), where we concatenate a linear positional encoding at the second-to-last layer. The resulting grid of visual features is flattened into a set of vectors $\boldsymbol{h}_t = f_{\text{enc}}(\boldsymbol{x}_t) \in \mathbb{R}^{N \times D_{\text{enc}}}$, where $N$ is the size of the flattened grid (i.e., width*height) and $D_{\text{enc}}$ is the dimensionality of the CNN feature maps. Afterwards, each vector is independently passed through a multi-layer perceptron (MLP).

**Slot Initialization** SAVi maintains $K$ slots, each of which can represent a part of the input such as objects or parts thereof. We denote the set of slot representations at time $t$ as $\mathcal{S}_t = [\boldsymbol{s}_t^1, \dots, \boldsymbol{s}_t^K] \in \mathbb{R}^{K \times D}$, where we use the calligraphic font to indicate that any operation on these sets is equivariant (or invariant) w.r.t. permutation of their elements. In other words, the ordering of the slots carries no information and they can be freely permuted (in a consistent manner across all time steps) without changing the model output. We consider two types of initializers: conditional and unconditional. In the conditional case, we encode the conditional input either via a simple MLP (in the case of bounding boxes or center of mass coordinates) or via a CNN (in the case of segmentation masks). For slots for which there is no conditioning information available (e.g. if $K$ is larger than the number of objects), we set the conditional input to a fixed value (e.g., '$-1$' for bounding box coordinates). The encoded conditional input forms the initial slot representation for SAVi. In the unconditional case, we either randomly initialize slots by sampling from a Gaussian distribution independently for each video (both at training and at test time), or by learning a set of initial slot vectors.

**Corrector** The task of the corrector is to update the slot representations based on the visual features from the encoder. In SAVi this is done using the iterative attention mechanism introduced in Slot Attention (Locatello et al., 2020). Different from a regular cross-attention mechanism (e.g. Vaswani et al., 2017) which is normalized over the inputs, Slot Attention encourages decomposition of the input into multiple slots via softmax-normalization over the output (i.e. the slots), which makes it an appealing choice for our video decomposition architecture. When using a single iteration of Slot Attention, the corrector update takes the following form:

$$\mathcal{U}_t = \frac{1}{\mathcal{Z}_t} \sum_{n=1}^{N} \mathcal{A}_{t,n} \odot v(\boldsymbol{h}_{t,n}) \in \mathbb{R}^{K \times D}, \quad \mathcal{A}_t = \underset{K}{\text{softmax}} \left( \frac{1}{\sqrt{D}} k(\boldsymbol{h}_t) \cdot q(\mathcal{S}_t)^T \right) \in \mathbb{R}^{N \times K}, \quad (1)$$

where $\mathcal{Z}_t = \sum_{n=1}^{N} \mathcal{A}_{t,n}$ and $\odot$ denotes the Hadamard product. $k$, $q$, $v$ are learned linear projections that map to a common dimension $D$. We apply LayerNorm (Ba et al., 2016) before each projection. The slot representations are then individually updated using Gated Recurrent Units (Cho et al., 2014) as $\hat{\boldsymbol{s}}_t^k = \text{GRU}(\boldsymbol{u}_t^k, \boldsymbol{s}_t^k)$. Alternatively, the attention step (followed by the GRU update) can be iterated multiple times with shared parameters per frame of the input video. For added expressiveness, we apply an MLP with residual connection $\hat{\boldsymbol{s}}_t^k \leftarrow \hat{\boldsymbol{s}}_t^k + \text{MLP}(\text{LN}(\hat{\boldsymbol{s}}_t^k))$ and LayerNorm (LN) after the GRU when using multiple Slot Attention iterations, following Locatello et al. (2020).

**Predictor** The predictor takes the role of a transition function to model temporal dynamics, including interactions between slots. To preserve permutation equivariance, we use a Transformer encoder (Vaswani et al., 2017). It allows for modeling of independent object dynamics as well as information exchange between slots via self-attention, while being more memory efficient than GNN-based models such as the Interaction Network (Battaglia et al., 2016). Slots are updated as follows:

$$\mathcal{S}_{t+1} = \text{LN}\big(\text{MLP}\big(\tilde{\mathcal{S}}_t\big) + \tilde{\mathcal{S}}_t\big), \quad \tilde{\mathcal{S}}_t = \text{LN}\big(\text{MultiHeadSelfAttn}\big(\hat{\mathcal{S}}_t\big) + \hat{\mathcal{S}}_t\big). \quad (2)$$

For MultiHeadSelfAttn we use the default multi-head dot-product attention mechanism from Vaswani et al. (2017). We apply LayerNorm (LN) after each residual connection.

**Decoder** The network output should be permutation equivariant (for per-slot outputs) or invariant (for global outputs) with respect to the slots. Slots can be read out either after application of the corrector or after the predictor (transition model). We decode slot representations after application of the corrector using a slot-wise Spatial Broadcast Decoder (Watters et al., 2019) to produce per-slot RGB predictions of the optical flow (or reconstructed frame) and an alpha mask. The alpha mask is normalized across slots via a softmax and used to perform a weighted sum over the slot-wise RGB reconstruction to arrive at a combined reconstructed frame:

$$\boldsymbol{y_t} = \sum_{k=1}^{K} \boldsymbol{m}_t^k \odot \boldsymbol{y}_t^k, \qquad \boldsymbol{m}_t = \underset{K}{\text{softmax}}\big(\hat{\boldsymbol{m}}_t^k\big), \qquad \hat{\boldsymbol{m}}_t^k, \boldsymbol{y}_t^k = f_{\text{dec}}\big(\hat{\boldsymbol{s}}_t^k\big). \quad (3)$$

**Training** Our sole prediction target is optical flow for each individual video frame, which we represent as RGB images using the default conversion in the literature (Sun et al., 2018). Alternatively,

our framework also supports prediction of other image-shaped targets, such as reconstruction of the original input frame. We minimize the pixel-wise squared reconstruction error (averaged over the batch), summed over both the temporal and spatial dimensions:

$$\mathcal{L}_{\text{rec}} = \sum_{t=1}^{T} \|\boldsymbol{y}_t - \boldsymbol{y}_t^{\text{true}}\|^2. \tag{4}$$

## 3 RELATED WORK

**Object-centric representation learning** There is a rich literature on learning object representations from static scenes (Greff et al., 2016; Eslami et al., 2016; Greff et al., 2017; 2019; Burgess et al., 2019; Engelcke et al., 2020; Crawford & Pineau, 2019; Lin et al., 2020; Locatello et al., 2020; Du et al., 2021a) or videos (van Steenkiste et al., 2018; Kosiorek et al., 2018; Stelzner et al., 2019; Kipf et al., 2020; Crawford & Pineau, 2020; Creswell et al., 2021) without explicit supervision. PSGNet (Bear et al., 2020) learns to decompose static images or individual frames from a video into hierarchical scene graphs using motion information estimated from neighboring video frames. Most closely related to our work are sequential object-centric models for videos and dynamic environments, such as OP3 (Veerapaneni et al., 2020), R-SQAIR (Stanić & Schmidhuber, 2019), ViMON (Weis et al., 2020), and SCALOR (Jiang et al., 2020), which learn an internal motion model for each object. SIMONe (Kabra et al., 2021) auto-encodes an entire video in parallel and learns temporally-abstracted representations of objects. OP3 (Veerapaneni et al., 2020) uses the same decoder as SAVi and a related dynamics model, but a less efficient inference process compared to Slot Attention.

In an attempt to bridge the gap to visually richer and more realistic environments, recent works in object-centric representation learning have explored integration of inductive biases related to 3D scene geometry, both for static scenes (Chen et al., 2020; Stelzner et al., 2021) and for videos (Du et al., 2021b; Henderson & Lampert, 2020; Harley et al., 2021). This is largely orthogonal to our approach of utilizing conditioning and optical flow. A recent related method, FlowCaps (Sabour et al., 2020), similarly proposed to use optical flow in a multi-object model. FlowCaps uses capsules (Sabour et al., 2017; Hinton et al., 2018) instead of a slot-based representation and assumes specialization of individual capsules to objects or parts of a certain appearance, making it unsuitable for environments that contain a large variety of object types. Using a slot-based, exchangeable representation of objects allows SAVi to represent a diverse range of objects and generalize to novel objects at test time.

We discuss further recent related works on attention-based architectures operating on sets of latent variables (Santoro et al., 2018; Goyal et al., 2021a;b;c; Jaegle et al., 2021b;a), object-centric models for dynamic visual reasoning (Yi et al., 2020; Ding et al., 2021a; Bar et al., 2021), and supervised attention-based object-centric models (Fuchs et al., 2019; Carion et al., 2020; Kamath et al., 2021; Meinhardt et al., 2021) in Appendix A.1.

**Video object segmentation and tracking** The conditional tasks we consider in our work are closely related to the computer vision task of semi-supervised video object segmentation (VOS), where segmentation masks are provided for the first video frame during evaluation. Different from the typical setting, which is addressed by supervised learning on fully annotated videos or related datasets (e.g. Caelles et al., 2017; Luiten et al., 2018), we consider the problem where models do not have access to any supervised information beyond the conditioning information on the first frame (e.g. a bounding box for each object). Several recent works have explored pre-training using self-supervision (Li et al., 2019; Jabri et al., 2020; Caron et al., 2021) or image classification (Zhang et al., 2020) for the semi-supervised VOS task. These models rely on having access to segmentation masks in the first frame at evaluation time. We demonstrate that multi-object segmentation and tracking can emerge even when segmentation labels are absent at both training and test time.

There is a rich literature on using motion cues to segment objects in the computer vision community. These methods use motion information at test time to, for example, cluster trajectories in order to segment independently moving objects (Faktor & Irani, 2014) or estimate multiple fundamental matrices between two views (Isack & Boykov, 2012), to name a few. Closest to our work is a contemporary method (Yang et al., 2021) that trains a Slot Attention model on isolated optical flow data for foreground-background segmentation of a single object, independently for individual video frames and without using visual observations. Our method, on the other hand, supports multi-object environments and only relies on motion information as a training signal but otherwise operates

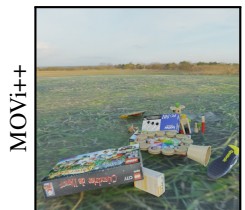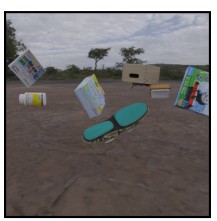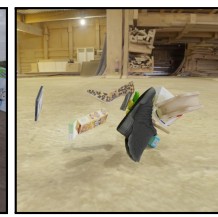

Figure 2: Frame samples from the multi-object video datasets used in our experiments. MOVi++ uses real-world backgrounds and 3D scanned objects, which is a significant step up in visual complexity compared to datasets used in prior work on unsupervised/weakly-supervised object-centric learning.

directly on textured visual information, which allows it to segment static scenes at test time where optical flow is unavailable and consistently represent and track multiple objects throughout a video.

## 4 EXPERIMENTS

We investigate 1) how SAVi compares to existing unsupervised video decomposition methods, 2) how various forms of hints (e.g., bounding boxes) can facilitate scene decomposition, and 3) how SAVi generalizes to unseen objects, backgrounds, and longer videos at test time.

**Metrics**   We report two metrics to measure the quality of video decomposition, object segmentation, and tracking: Adjusted Rand Index (ARI) and mean Intersection over Union (mIoU). ARI is a clustering similarity metric which we use to measure how well predicted segmentation masks match ground-truth masks in a permutation-invariant fashion, which makes it suitable for unsupervised methods. Like in prior work (Greff et al., 2019; Locatello et al., 2020; Kabra et al., 2021), we only measure ARI based on foreground objects, which we refer to as FG-ARI. For video data, one cluster in the computation of ARI corresponds to the segmentation of a single object for the entire video, requiring temporal consistency without object identity switches to perform well on this metric.

**Training setup**   During training, we split each video into consecutive sub-sequences of 6 frames each, where we provide the conditioning signal for the first frame. We train for 100k steps (200k for fully unsupervised video decomposition) with a batch size of 64 using Adam (Kingma & Ba, 2015) with a base learning rate of $2 \cdot 10^{-4}$. We use a total of 11 slots in SAVi. For our experiments on fully unsupervised video decomposition we use 2 iterations of Slot Attention per frame and a single iteration otherwise. Further architecture details and hyperparameters are provided in the appendix. On 8x V100 GPUs with 16GB memory each, training SAVi with bounding box conditioning takes approx. 12hrs for videos with $64 \times 64$ resolution and 30hrs for videos with $128 \times 128$ resolution.

### 4.1 UNSUPERVISED VIDEO DECOMPOSITION

We first evaluate SAVi in the unconditional setting and with a standard RGB reconstruction objective on the CATER (Girdhar & Ramanan, 2019) dataset[1]. Dataset examples are shown in Figure 2. Results of SAVi and four unsupervised object representation learning baselines (taken from Kabra et al. (2021)) are summarized in Table 1a. The two image-based methods (Slot Attention (Locatello et al., 2020) and MONet (Burgess et al., 2019)) are independently applied to each frame, and thus lack a built-in notion of temporal consistency which is reflected in their poor FG-ARI scores. Unsurprisingly, the two video-based baselines, S-IODINE (Greff et al., 2019) and SIMONe (Kabra et al., 2021), perform better. The (unconditional) SAVi model outperforms these baselines, demonstrating the adequacy of our architecture for the task of unsupervised object representation learning, albeit only for the case of simple synthetic data. For qualitative results, see Figure 5c and Appendix A.3.

### 4.2 MORE REALISTIC DATASETS

To test SAVi on more realistic data, we use two video datasets (see Figure 2) introduced in Kubric (Greff et al., 2021), created by simulating rigid body dynamics of up to ten 3D objects rendered in various scenes via raytracing using Blender (Blender Online Community, 2021). We refer to these as

---

[1]We use a variant of CATER with segmentation masks (for evaluation), introduced by Kabra et al. (2021).

Table 1: Segmentation results. Mean ± standard error (5 seeds). All values in %.

(a) Unsupervised.

| Model | CATER FG-ARI↑ |
|---|---|
| Slot Attention | $7.3 \pm 0.3$ |
| MONet | $41.2 \pm 0.5$ |
| S-IODINE | $66.8 \pm 1.5$ |
| SIMONe | $91.8 \pm 1.6$ |
| SAVi (uncond.) | $92.8 \pm 0.8$ |
| SAVi + Segm. | $97.9 \pm 0.4$ |

(b) Conditional with optical flow supervision.

| Model (+ Conditioning) | MOVi | | MOVi++ | |
|---|---|---|---|---|
| | FG-ARI↑ | mIoU↑ | FG-ARI↑ | mIoU↑ |
| Segmentation Propagation | $61.5 \pm 0.6$ | $49.8 \pm 0.4$ | $37.8 \pm 0.5$ | $33.3 \pm 0.2$ |
| SAVi + Segmentation | $\mathbf{93.7} \pm 0.2$ | $\mathbf{72.0} \pm 0.3$ | $70.4 \pm 2.0$ | $43.0 \pm 0.6$ |
| SAVi + Bounding Box | $\mathbf{93.7} \pm 0.0$ | $71.2 \pm 0.6$ | $77.4 \pm 0.5$ | $\mathbf{45.9} \pm 1.2$ |
| SAVi + Center of Mass | $\mathbf{93.8} \pm 0.1$ | $\mathbf{72.1} \pm 0.2$ | $78.3 \pm 0.6$ | $43.5 \pm 2.9$ |
| CRW (Jabri et al., 2020) | – | 42.4 | – | 50.9 |
| T-VOS (Zhang et al., 2020) | – | 50.4 | – | 46.4 |
| SAVi (ResNet) + Bounding Box | – | – | $82.8 \pm 0.4$ | $50.7 \pm 0.2$ |
| Flow k-Means + Center of Mass | 30.2 | 5.8 | 32.6 | 9.3 |

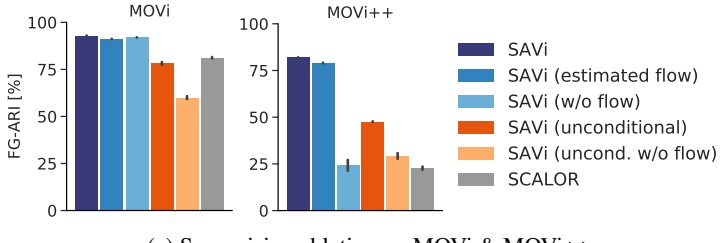

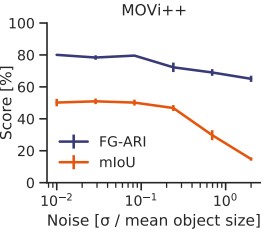

(a) Supervision ablation on MOVi & MOVi++.      (b) Robustness to noise.

Figure 3: (**a**) Supervision ablations for a SAVi model trained with bounding boxes as conditional input. (**b**) Robustness analysis with Gaussian noise added to the conditioning signal. In both figures, we evaluate on the first six frames and report mean ± standard error across 5 seeds.

Multi-Object Video (MOVi) datasets. The first dataset, MOVi, uses the same simple 3D shapes as the CATER dataset (both inspired by the CLEVR benchmark (Johnson et al., 2017)), but introduces more complex rigid-body physical dynamics with frequent collisions and occlusions. The second dataset, MOVi++ significantly increases the visual complexity compared to datasets used in prior work and uses approx. 380 high resolution HDR photos as backgrounds and a set of 1028 3D scanned everyday objects (Google Research, 2020), such as shoes, toys, or household equipment. Each dataset contains 9000 training videos and 1000 validation videos with 24 frames at 12 fps each and, unless otherwise mentioned, a resolution of $64 \times 64$ pixels for MOVi and $128 \times 128$ pixels for MOVi++.

Due to its high visual complexity the MOVi++ dataset is a significantly more challenging testcase for unsupervised and weakly-supervised instance segmentation. In fact, while SCALOR (Jiang et al., 2020) (using the reference implementation) was able to achieve $81.2 \pm 0.4\%$ FG-ARI on MOVi, it only resulted in $22.7 \pm 0.9\%$ FG-ARI on MOVi++. The SIMONe model also only achieved around 33% FG-ARI (with one outlier as high as 46%), using our own re-implementation (which approximately reproduces the above results on CATER). We observed that both models, as well as SAVi, converged to a degenerate solution where slots primarily represent fixed regions of the image (often a rectangular patch or a stripe), instead of tracking individual objects.

### 4.3 CONDITIONAL VIDEO DECOMPOSITION

To tackle these more realistic videos we 1) change the training objective from predicting RGB image to optical flow and 2) condition the latent slots of the SAVi model on *hints* about objects in the first frame of the video, such as their segmentation mask. We showcase typical qualitative results for SAVi in Figure 4, and on especially challenging cases with heavy occlusion or complex textures in Figures 5a and 5b. In all cases, SAVi was trained on six consecutive frames, but we show results for full (24 frame) sequences at test time. Using detailed segmentation information for the first frame is common practice in video object segmentation, where models such as T-VOS (Zhang et al., 2020) or CRW (Jabri et al., 2020) *propagate* the initial masks through the remainder of the video sequence.

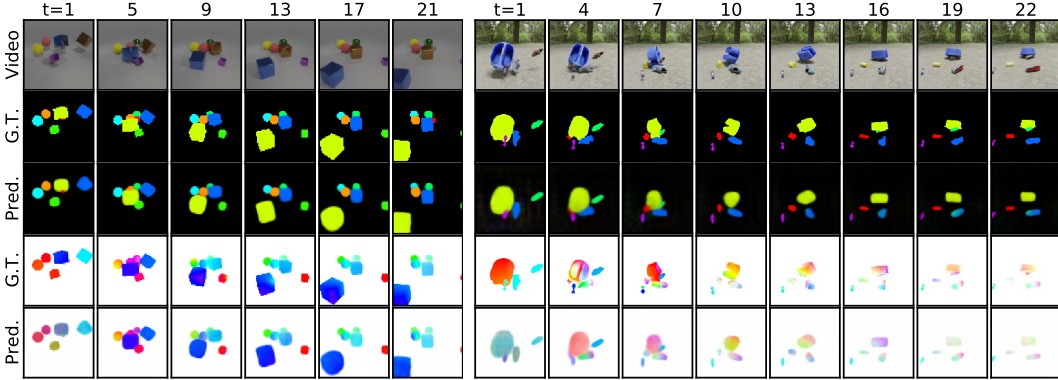

Figure 4: Representative qualitative results of SAVi conditioned on bounding box annotations in the first video frame. We visualize predicted (*Pred.*) segmentations and optical flow predictions together with their respective ground-truth values (*G.T.*) for both MOVi (*left*) and MOVi++ (*right*). We visualize the soft segmentation masks provided by the decoder of SAVi. Each mask is multiplied with a color that identifies the particular slot it represents.

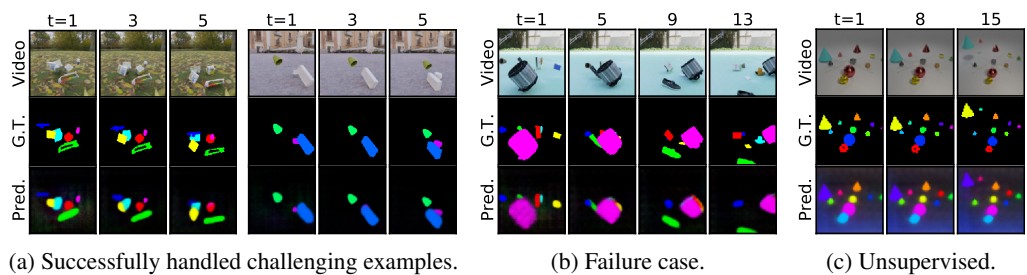

(a) Successfully handled challenging examples.    (b) Failure case.    (c) Unsupervised.

Figure 5: (**a**) SAVi successfully handles examples with complex texture (*left*) and occlusion with object of similar appearance (*right*). (**b**) In a heavily occluded example SAVi can lose track of objects and swap object identities. (**c**) Unsupervised segmentation on CATER with RGB reconstruction.

**Quantitative Evaluation** On the MOVi and MOVi++ datasets T-VOS achieves $50.4\%$ and $46.4\%$ mIoU respectively, whereas CRW achieves $42.4\%$ and $50.9\%$ mIoU (see Table 1b). SAVi learns to produce temporally consistent masks that are significantly better on MOVi ($72.0\%$ mIoU) and slightly worse than T-VOS and CRW on MOVi++ ($43.0\%$ mIoU) when trained to predict flow and with segmentation masks as conditioning signal in the first frame. However, we hold that this comparison is not entirely fair, since both CRW and T-VOS use a more expressive ResNet (He et al., 2016) backbone and operate on higher-resolution frames (480p). For T-VOS, this backbone is pre-trained on ImageNet (Deng et al., 2009), whereas for CRW we re-train it on MOVi/MOVi++. For a fairer comparison, we adapt T-VOS (Zhang et al., 2020) to our setting by using the same CNN encoder architecture at the same resolution as in SAVi and trained using the same supervision signal (i.e. optical flow of moving objects). While this adapted baseline (*Segmentation Propagation* in Table 1b) performs similar to T-VOS on MOVi, there is a significant drop in performance on MOVi++. We verify that SAVi can similarly benefit from using a stronger ResNet backbone on MOVi++ (*SAVi (ResNet)* in Table 1b), which closes the gap to the CRW (Jabri et al., 2020) baseline.

**Conditioning Ablations** Surprisingly, we find that even simple hints about objects in the first video frame such as bounding boxes, or even just a point in the center of a bounding box (termed as *center of mass*), suffice for the model to establish correspondence between hint and visual input, and to retain the same quality in object segmentation and tracking. Conditioning on such simple signals is not possible with segmentation propagation models (Zhang et al., 2020; Jabri et al., 2020) as they require accurate segmentation masks as input. To study whether an even weaker conditioning signal suffices, we add noise — sampled from a Gaussian distribution $\mathcal{N}(0, \sigma)$ — to the center-of-mass coordinates provided to the model in the first frame, both during training and testing. We find that SAVi remains robust to noise scales $\sigma$ up to $\sim 20\%$ of the average object size (Figure 3b), after which point mIoU starts to decay (and faster than FG-ARI), indicating that SAVi starts to lose correspondence between

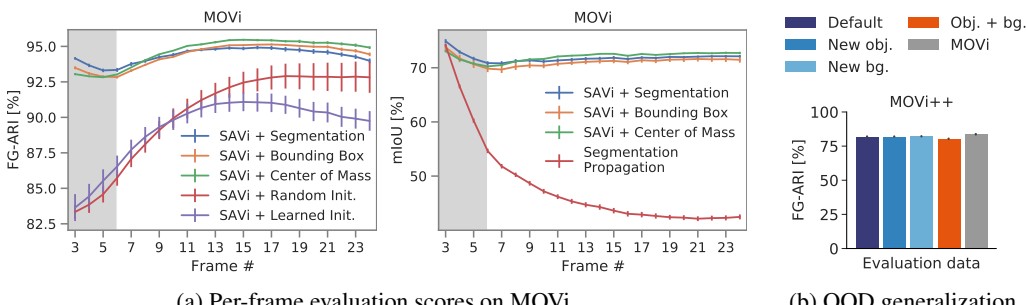

(a) Per-frame evaluation scores on MOVi.        (b) OOD generalization.

Figure 6: (**a**) Per-frame segmentation quality. Grey area highlights the number of frames used during training. (**b**) Video decomposition results on out-of-distribution evaluation splits with new objects and/or new backgrounds (6 frames). Mean $\pm$ standard error for 5 seeds.

the conditioning signal and the objects in the scene. Full ablation of the conditioning signal leads to a significant drop in performance (Figure 3), which demonstrates that conditional hints are indeed important to SAVi, though even very rough hints suffice.

**Optical flow ablations**     Using ground-truth optical flow as a training target — especially in the absence of camera motion — is a form of weak supervision that won't be available for real data. To investigate its benefits and importance, we first apply k-means clustering to the optical flow signal (enriched with pixel position information) where we use the center of mass coordinates of objects to initialize the cluster centers. The results (see Table 1b) show that simply clustering the optical flow signal itself is insufficient for accurate segmentation. Next, we perform the following ablations on a SAVi model with bounding boxes as conditional input, summarized in Figure 3: 1) instead of using ground-truth optical flow provided by the simulator/renderer, we use approximate optical flow obtained from a recent unsupervised flow estimation model (Stone et al., 2021) (*estimated flow*); 2) we replace optical flow with RGB pixel values as training targets, i.e. using a reconstruction target (*w/o flow*); 3) we train SAVi without conditioning by initializing slots using fixed parameters that are learned during training (*unconditional*). We find, that estimated flow confers virtually the same benefits as ground truth-flow, and that flow is unnecessary for the simpler MOVi dataset. For MOVi++, on the other hand, it is crucial for learning an accurate decomposition of the scene, and without access to optical flow (true or estimated) SAVi fails to learn meaningful object segmentations.

### 4.4 Test time generalization

**Longer sequences**     For memory efficiency, we train SAVi on subsequences with a total of 6 frames, i.e. shorter than the full video length. We find, however, that even without any additional data augmentation or regularization, SAVi generalizes well to longer sequences at test time, far beyond the setting used in training. We find that segmentation quality measured in terms of FG-ARI remains largely stable, and surprisingly even increases for unconditional models (Figure 6a). Part of the increase can be explained by objects leaving the field of view later in the video (see examples in the appendix), but models nonetheless appear to benefit from additional time steps to reliably break symmetry and bind objects to slots. The opposite effect can be observed for segmentation propagation models: performance typically decays over time as errors accumulate.

**New objects and backgrounds**     We find that SAVi generalizes surprisingly well to settings with previously unseen objects and with new backgrounds at test time: there is no significant difference in both FG-ARI and mIoU metrics when evaluating SAVi with bounding box conditioning on evaluation sets of MOVi++ with entirely new objects or backgrounds. On an evaluation set where we provide both new backrounds and new objects at the same time, there is a drop of less than 2% (absolute), see Figure 6b. Remarkably, a SAVi model trained on MOVi++ also transfers well to MOVi at test time.

**Part-whole segmentation**     Our dataset is annotated only at the level of whole objects, yet we find anecdotal evidence that by conditioning at the level of parts during inference the model can in fact successfully segment and track the corresponding parts. Consider, for example the two *green fists* in Figure 7 which in the dataset are treated as a single composite object. When conditioned with a

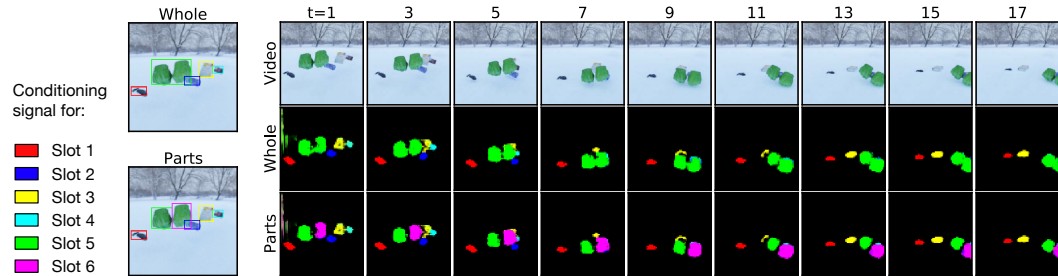

Figure 7: Emergent part-whole segmentation by querying the model at test time in different ways: The model attends to either both *green fist* with a single slot or to each individual fist with a separate slot, depending on the granularity of the conditioning signal. We visualize SAVi's attention masks.

single bounding box encompassing both fists (*Whole*), or on two bounding boxes (*Parts*), SAVi either segments and tracks either the composite object or the individual parts (see appendix for an additional example). This finding is remarkable for two reasons: Firstly, it shows that the model can (at least to some degree) generalize to different levels of granularity. This kind of generalization is very useful, since the notion of an object is generally task-dependent and the correct level of granularity might not be known at training time. Secondly, this experiments demonstrates a powerful means for interfacing with the model: rather than producing a single fixed decomposition, SAVi has learned to adapt its segmentation and processing of the video based on the conditioning signal. We believe that this way of learning a part-whole hierarchy only implicitly and segmenting different levels on demand is ultimately more flexible and tractable than trying to explicitly represent the entire hierarchy at once.

## 4.5 LIMITATIONS

There are still several obstacles that have to be overcome for successful application of our framework to the full visual and dynamic complexity of the real world. First, our training set up assumes the availability of optical flow information at training time, which may not be available in real-world videos. Our experiments, however, have demonstrated that estimated optical flow from an unsupervised model such as SMURF (Stone et al., 2021) can be sufficient to train SAVi. Secondly, the environments considered in our work, despite taking a large step towards realism compared to prior works, are limited by containing solely rigid objects with simple physics and, in terms of the MOVi datasets, solely moving objects. In this case, optical flow, even in the presence of camera movement, can be interpreted as an indirect supervision signal for foreground-background (but not per-object) segmentation. Furthermore, training solely with optical flow presents a challenge for static objects. Nonetheless, we find evidence that SAVi can reliably decompose videos in a constrained real-world environment: we provide qualitative results demonstrating scene decomposition and long-term tracking in a robotic grasping environment (Cabi et al., 2020) in Appendix A.3. Most real-world video data, however, is still of significantly higher complexity than the environments considered here and reliably bridging this gap is an open problem.

## 5 CONCLUSION

We have looked at the problem of learning object representations and physical dynamics from video. We introduced SAVi, an object-centric architecture that uses attention over a latent set of slots to discover, represent and temporally track individual entities in the input, and we have shown that video in general and flow prediction in particular are helpful for identifying and tracking objects. We demonstrated that providing the model with simple and possibly unreliable position information about the objects in the first frame is sufficient to steer learning towards the right level of granularity to successfully decompose complex videos and simultaneously track and segment multiple objects. More importantly, we show that the level of granularity can potentially be controlled at inference time, allowing a trained model to track either parts of objects or entire objects depending on the given initialization. Since our model achieves very reasonable segmentation and tracking performance, this is evidence that object-centric representation learning is not primarily limited by model capacity. This way of conditioning the initialization of slot representations with location information also potentially opens the door for a wide range of semi-supervised approaches.

ETHICS STATEMENT

Our analysis, which is focused on multi-object video data with physically simulated common household objects, has no immediate impact on general society. As with any model that performs dynamic scene understanding, applications with potential negative societal impact such as in the area of surveillance, derived from future research in this area, cannot be fully excluded. We anticipate that future work will take steps to close the gap between the types of environments considered in this work, and videos of diverse scenes taken in the real world, which will likely enable applications in various societally relevant domains such as robotics and general video understanding.

ACKNOWLEDGEMENTS

We would like to thank Mike Mozer and Daniel Keysers for general advice and feedback on the paper, and David Fleet, Dirk Weissenborn, Jakob Uszkoreit, Marvin Ritter, Andreas Steiner, and Etienne Pot for helpful discussions. We are further grateful to Rishabh Kabra for sharing the CATER (with masks) dataset and to Yi Yang, Misha Denil, Yusuf Aytar, and Claudio Fantacci for helping us get started with the Sketchy dataset.

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

# A    APPENDIX

In Section A.1, we discuss additional related work. In Sections A.2–A.4, we show additional qualitative and quantitative results. Section A.4 further contains results for an ablation study on SAVi. Lastly, in Sections A.5–A.8, we report additional details on datasets, model architecture, and training setup (hyperparameters, baselines, metrics).

## A.1    ADDITIONAL RELATED WORK

**Attention-based networks for sets of latent variables**    SAVi shares close connections with other attention-based modular neural networks. Relational Recurrent Networks (Santoro et al., 2018) and the RIM model (Goyal et al., 2021c) use related attention mechanisms to map from inputs to slots and from slots to slots, the latter of which (RIM) uses specialized neural network parameters for each slot. Follow-up works (Goyal et al., 2021b;a) introduce additional sparsity and factorization constraints on the dynamics model, which is orthogonal to our contributions of conditional learning and dynamics modeling using optical flow. The Perceiver (Jaegle et al., 2021b) uses an architecture similar to Slot Attention (Locatello et al., 2020), i.e. using a set of latent vectors which are (recurrently) updated by attending on a set of input feature vectors, for classification tasks. The Perceiver IO (Jaegle et al., 2021a) model extends this architecture by adding an attention-based decoder component to interface with a variety of structured prediction tasks. SAVi similarly maps inputs onto a latent set of slot vectors using an attention mechanism, but differs in the decoder architecture and in its sequential nature: different from Perceiver IO, SAVi processes a sequence of inputs (e.g. video frames) of arbitrary length, encoding and decoding a single input (or output) element per time step while maintaining a latent set of slot variables that is carried forward in time.

**Supervised slot-based models for visual tracking**    Slot-based architectures have also been explored in the context of fully supervised multi-object tracking and segmentation with models such as MOHART (Fuchs et al., 2019), TrackFormer (Meinhardt et al., 2021), and TubeR (Zhao et al., 2021). In the image domain, GPV-I (Tanmay Gupta, 2021) and MDETR (Kamath et al., 2021; Carion et al., 2020) explore conditioning supervised slot-based models on auxiliary information, in a form which requires combinatorial matching during training. SAVi learns to track and segment multiple objects without being directly supervised to do so, using only weak forms of supervision for the first frame of the video. Slot conditioning allows us to avoid matching and to consistently identify and track individual objects by their specified query.

**Dynamic visual reasoning and action graphs**    Object-centric models for dynamic scenes have found success in other relational tasks in vision, such as supervised visual reasoning (Yi et al., 2020; Chen et al., 2021; Ding et al., 2021a;b), activity/action recognition (Jain et al., 2016; Girdhar et al., 2019; Herzig et al., 2019), and compositional video synthesis with action graphs (Bar et al., 2021). Many of these approaches use pre-trained (Chen et al., 2021) and sometimes frozen (Yi et al., 2020; Ding et al., 2021a) object detection backbones. Combining these methods with an end-to-end video object discovery architecture such as SAVi using self-supervised learning objectives is an interesting direction for future work.

## A.2    REAL-WORLD ROBOTICS TASK

To evaluate whether SAVi can decompose real-world video data in a limited domain, we train and qualitatively evaluate an unsupervised SAVi model with RGB reconstruction on the Sketchy dataset (Cabi et al., 2020). The Sketchy dataset contains videos of a real-world robotic grasper interacting with various objects.

We use the human demonstration sequences as part of the "rgb30_all" subset of the Sketchy dataset (Cabi et al., 2020), resulting in a total of 2930 training videos of 201 frames each. In Figure A.1 and in the supplementary videos (see supplementary material), we show qualitative results on unseen validation data.

Our qualitative results in Figure A.1 demonstrate that SAVi can decompose these real-world scenes into meaningful object components and consistently represent and track individual scene components over long time horizons, far beyond what is observed during training. SAVi is trained with a per-slot

MLP predictor model (with a single hidden layer of 256 hidden units, a skip connection, and Layer Normalization) for 1M steps with otherwise the same architecture as the unsupervised SAVi model used for CATER described in the main paper.

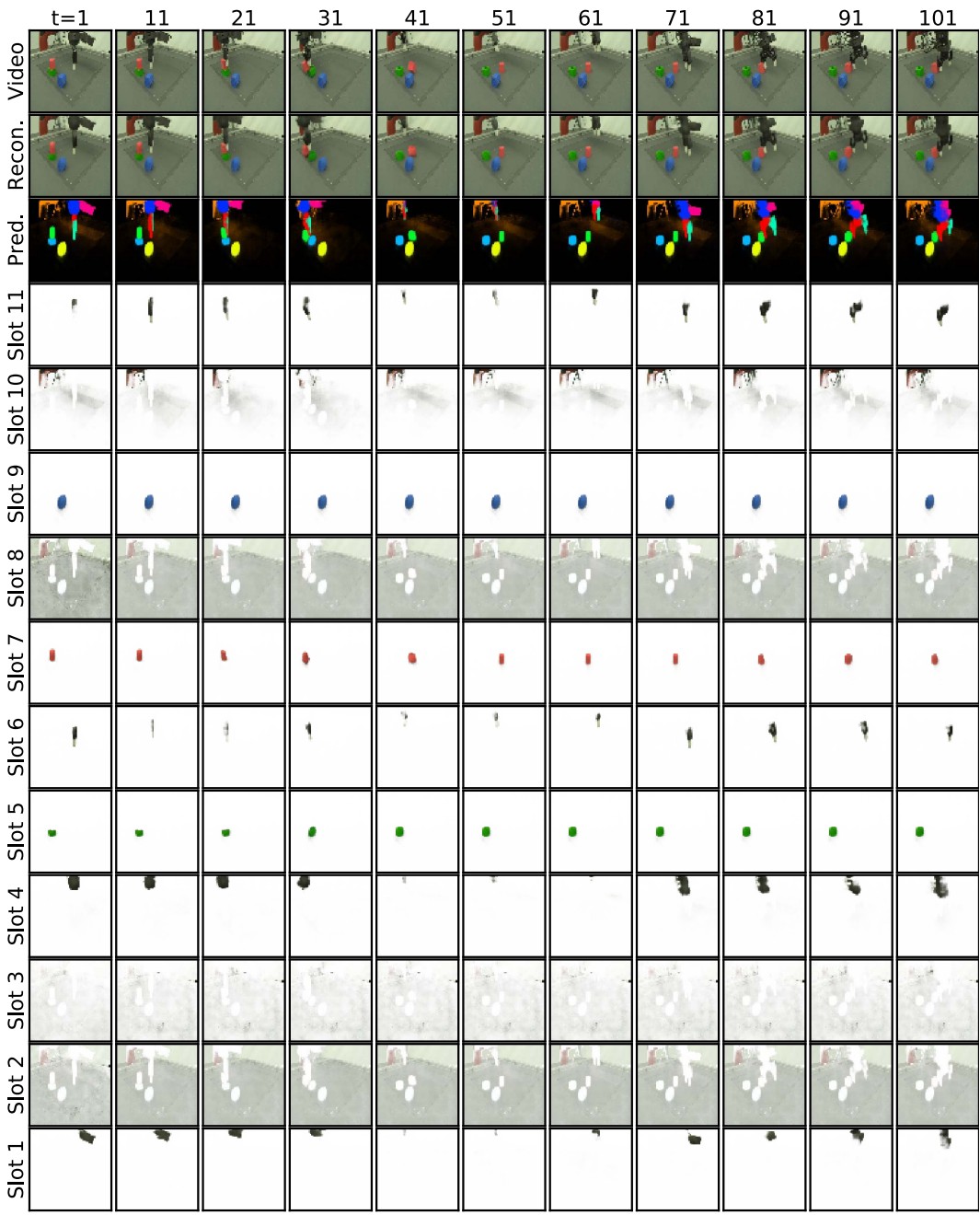

Figure A.1: Fully-unsupervised video decomposition on Sketchy (Cabi et al., 2020), a real-world robotic grasping dataset, with a SAVi model trained using 11 object slots. In addition to the predicted segmentation of the scene (*Pred.*), we visualize reconstructions of individual slots (multiplied with their respective predicted soft segmentation mask). We assign all three discovered background slots (slots 2, 3, and 8) the color black to ease interpretation. Additional videos are provided in the supplementary material.

## A.3 Additional qualitative results

**Part-whole segmentation** In Figure A.2 we show another example of steerable part-whole segmentation based on the granularity of the hint provided for the first frame. In this example, the laptop in the foreground of the scene is either annotated with a single bounding box that covers the full object (which is the granularity of annotation originally provided in the dataset), or we re-annotate this object as two separate instances, where one bounding box covers the screen of the laptop and the other bounding box covers the base incl. the keyboard. Despite never having seen a separate "screen" object in the dataset, the model can reliably identify, segment, and track each individual part, as can be seen both by the predicted decoder segmentation masks and the corrector attention masks. Alternatively, if the conditional input covers the full object, the model tracks and segments the entire laptop using a single slot, here shown in green.

We further observe that the corrector attention masks much more sharply outline the object compared to the decoder segmentation masks, but have artifacts for large objects which is likely due to the limited receptive field of the encoder CNN used in our work (for reasons of simplicity). Both this gap in sharpness and the artifacts seen in the corrector masks for large objects can likely be addressed in future work by using more powerful encoder and decoder architectures.

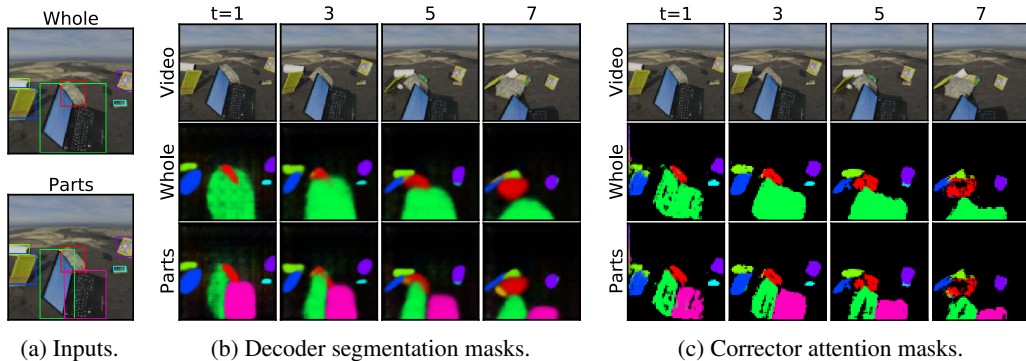

(a) Inputs.  (b) Decoder segmentation masks.  (c) Corrector attention masks.

Figure A.2: An example of query-dependent part vs. whole segmentation. (a) shows the input bounding boxes provided as conditioning signal. (b) and (c) show the per-slot segmentations obtained for each of the two conditioning cases in terms of the decoder segmentation masks and corrector attention masks, respectively.

**Unsupervised video decomposition** In Figure A.3 we show qualitative results for fully-unsupervised video decomposition with SAVi on the CATER dataset (on unseen validation data). Since SAVi is trained with a RGB reconstruction loss in this setting, we can visualize reconstructions of individual objects over multiple time steps in the video.

**Extrapolation to long sequences** In Figures A.4–A.5 we show several representative examples of applying SAVi with bounding box conditioning on full-length MOVi and MOVi++ videos of 24 frames, after being trained on sequences of only 6 frames.

**Moving camera** In Figure A.6 we show qualitative results for SAVi (with bounding box conditioning) on a more challenging variant of the MOVi++ dataset with (linear) camera movement, where optical flow encodes both object movement and movement of the background relative to the camera. This effectively leads to colored background in the optical flow images, and weakens the usefulness of optical flow for foreground-background segmentation. Nonetheless, SAVi with bounding box conditioning achieves approx. $65.5 \pm 0.5\%$ FG-ARI on this modified dataset, and we find that it is still able to segment and track objects, although at a lower fidelity than in MOVi++.

**Baselines: Segmentation Propagation** In Figure A.7 we show qualitative results for the Segmentation Propagation baseline on a couple of sequences each in the MOVi and MOVi++ datasets. In these image grids, 'Pred.' corresponds to the soft segmentation masks before argmax similar to the illustrations for SAVi above. We also show 'Pred. Argmax' which corresponds to the hard

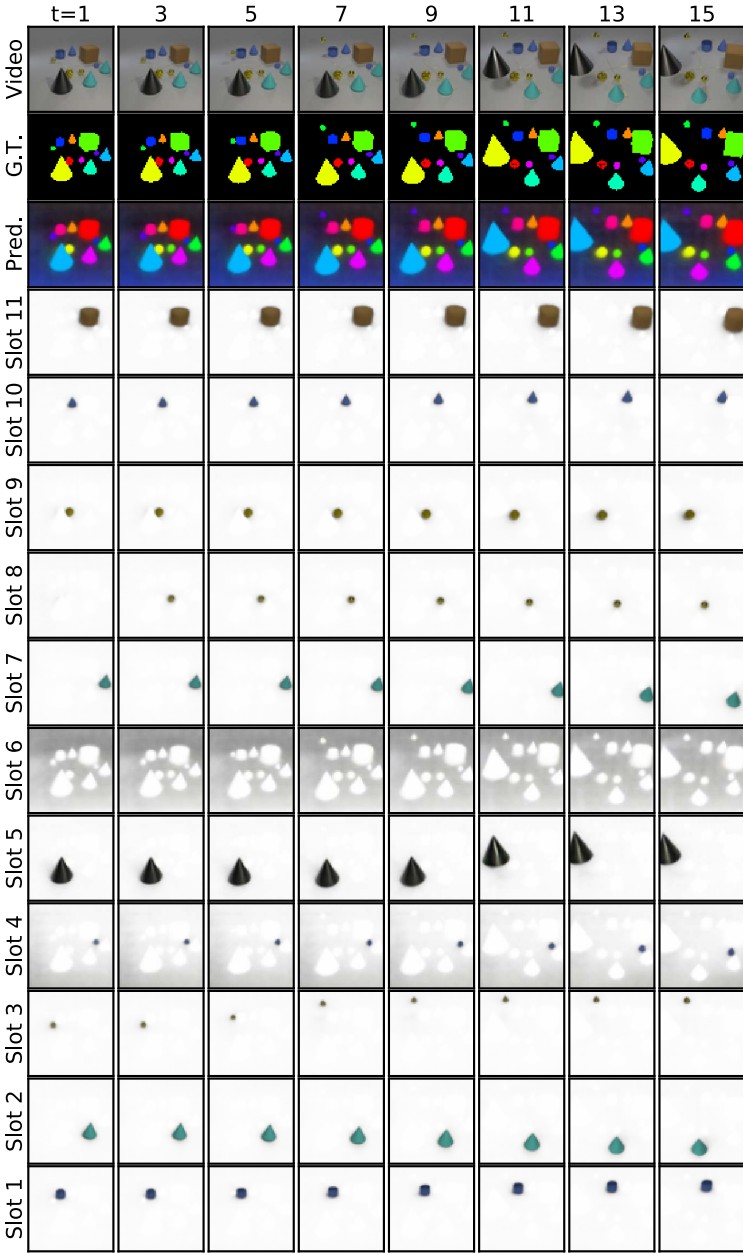

Figure A.3: Fully-unsupervised video decomposition on CATER with a SAVi model trained using 11 object slots. In addition to the predicted segmentation of the scene (*Pred.*), we visualize reconstructions of individual slots (multiplied with their respective predicted soft segmentation mask).

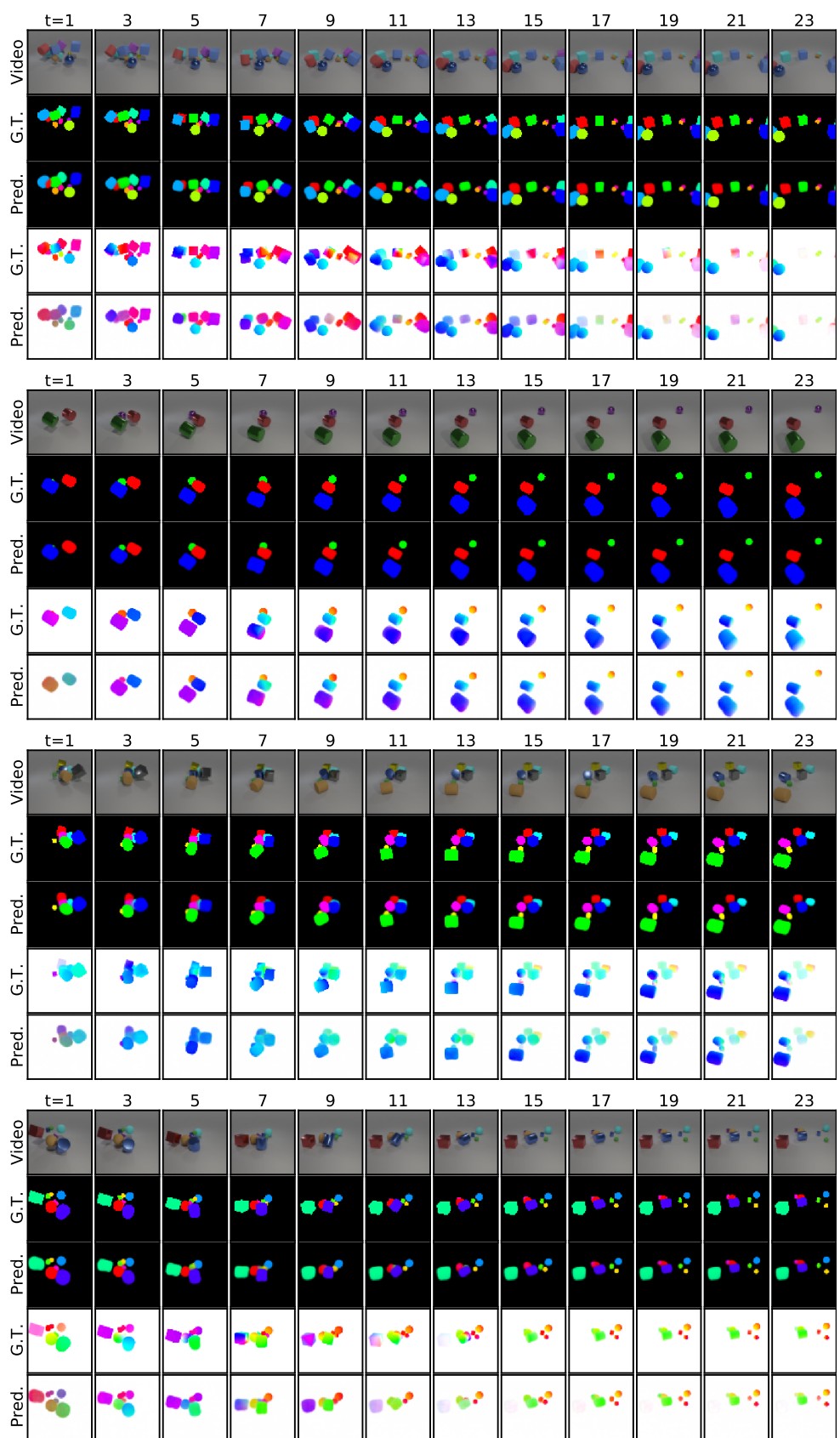

Figure A.4: Qualitative extrapolation results for a SAVi model with bounding box conditioning trained on sequences of 6 frames on MOVi.

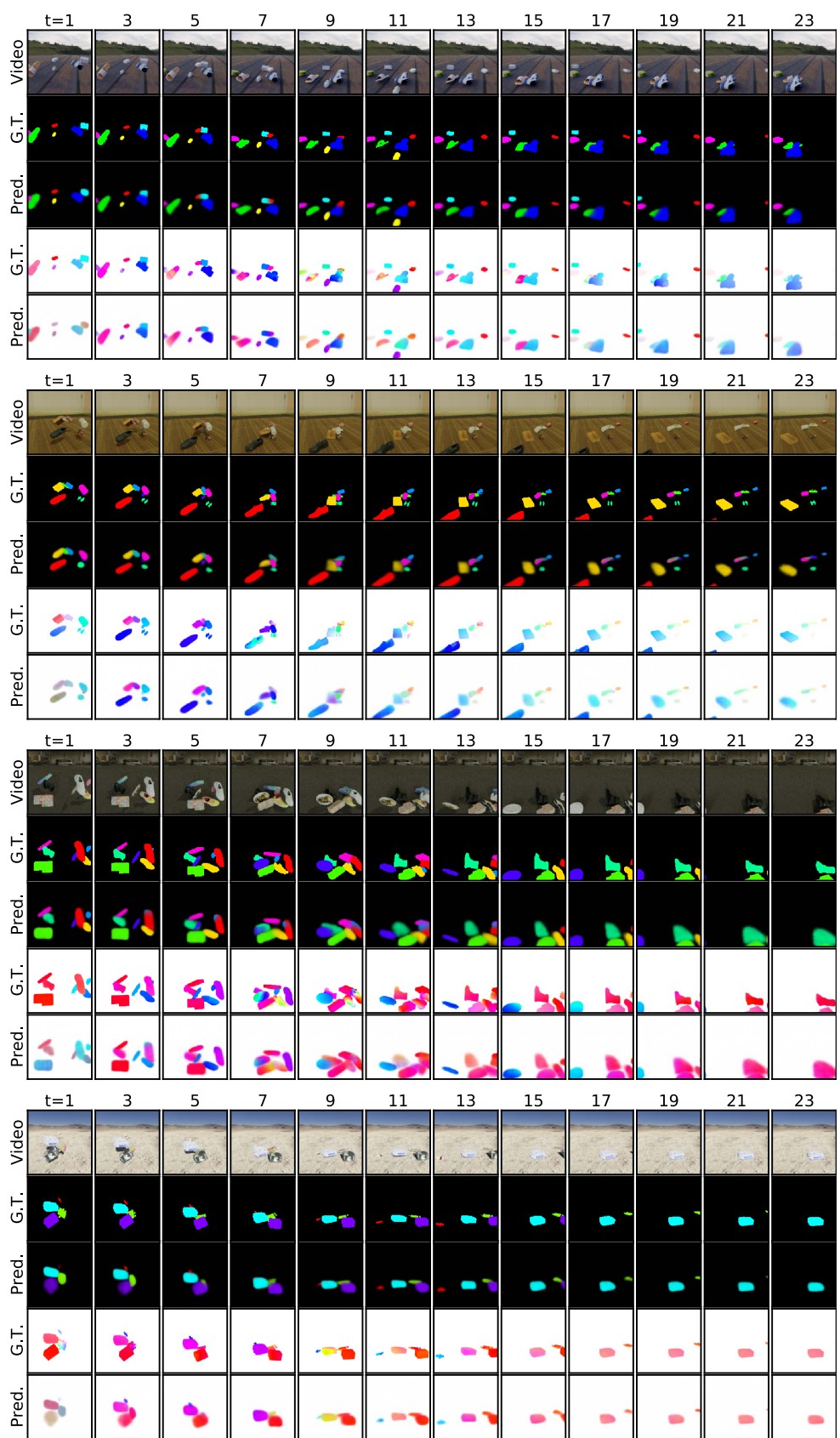

Figure A.5: Qualitative extrapolation results for a SAVi model with bounding box conditioning trained on sequences of 6 frames on MOVi++.

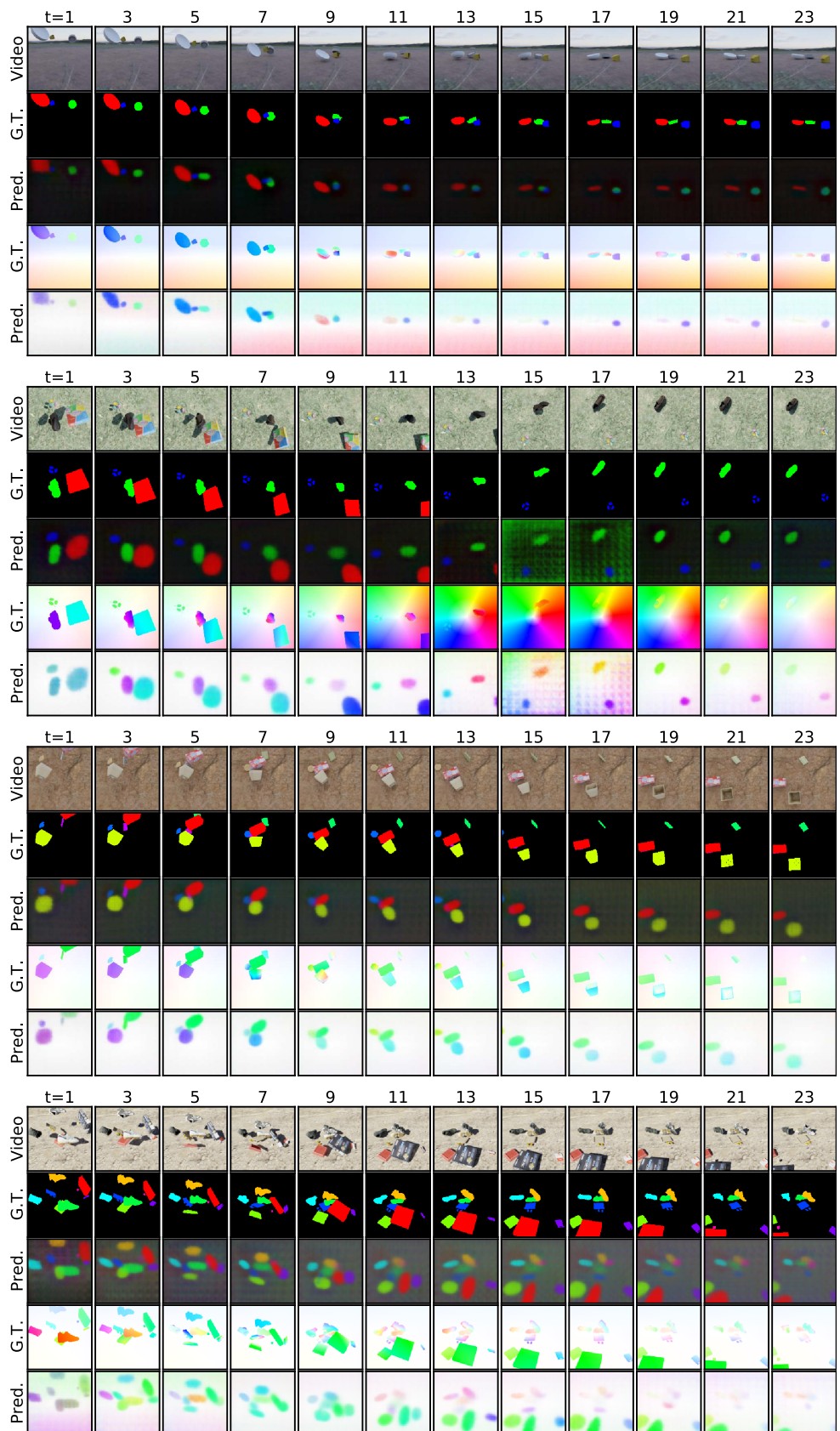

Figure A.6: Qualitative results for a SAVi model with bounding box conditioning trained on a varaint of MOVi++ with a moving camera.

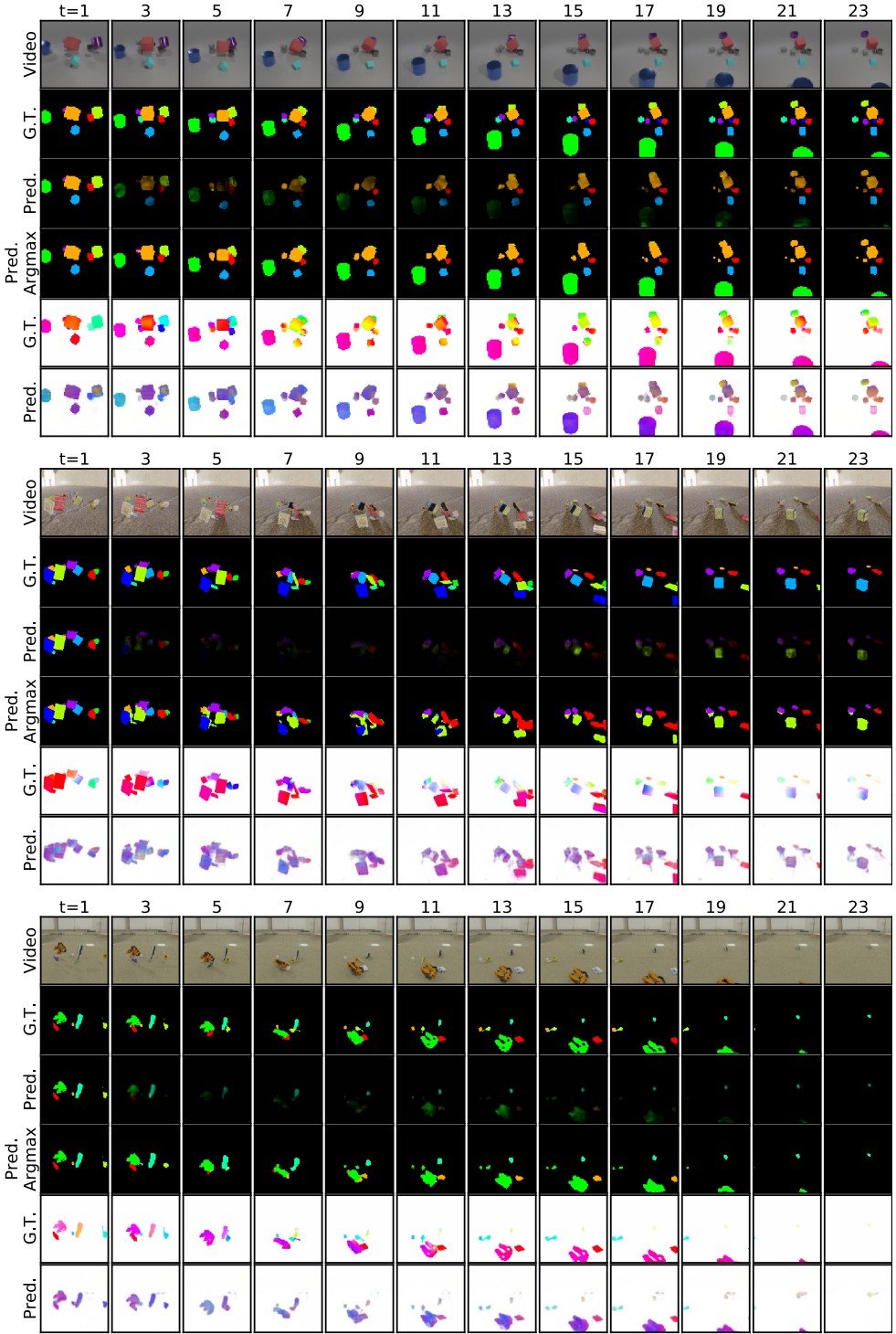

Figure A.7: Qualitative results for the Segmentation Propagation baseline with segmentation conditioning on the MOVi (top example) and on the MOVi++ (bottom two examples) datasets.

segmentation masks post argmax. The last row in each image grid corresponds to optical flow predicted from a single image. This is expected to be poor since flow prediction from a single image is a highly ambiguous task. We find that the Segmentation Propagation baseline struggles to retain small objects. These often disappear or become part of a bigger object. The soft masks also suggest that the model is struggling to keep the background separate from objects. 9 continuous reference frames were used for label propagation in Segmentation Propagation. See Section A.7 for more details.

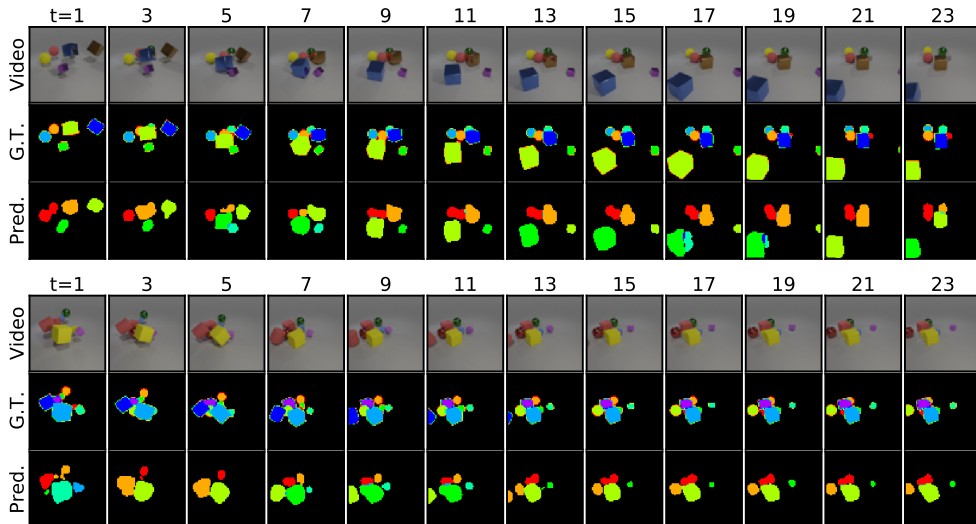

Figure A.8: Qualitative results for a SCALOR model trained on sequences of 6 frames on MOVi.

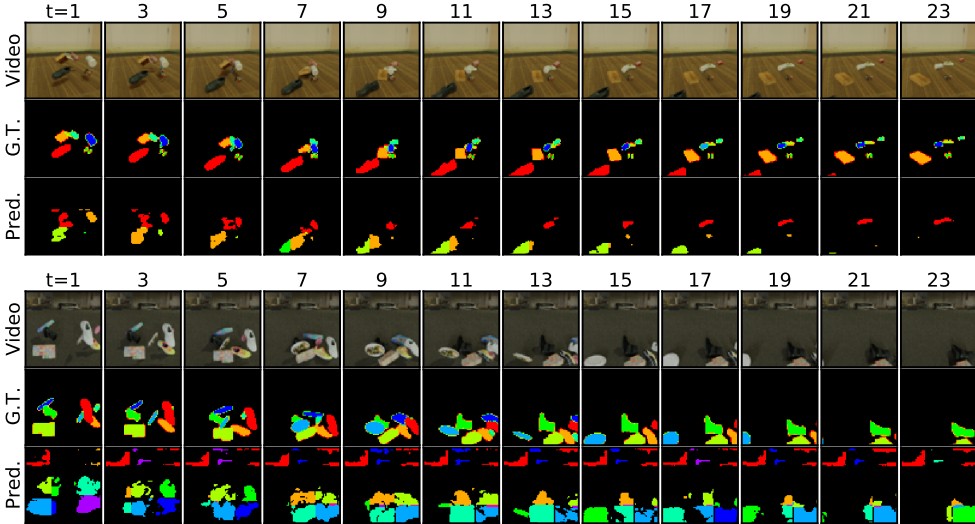

Figure A.9: Qualitative results for a SCALOR model trained on sequences of 6 frames on MOVi++.

**Baselines: SCALOR** Figures A.8 and A.9 show qualitative results for the SCALOR baseline on examples from the MOVi and MOVi++ datasets. Often, when small objects in MOVi get close to each other, SCALOR merges them as one object. On MOVi++, SCALOR struggles with object boundaries and often picks up background texture as extra objects.

## A.4 ADDITIONAL QUANTITATIVE RESULTS

**Transfer to static scenes** A key advantage of SAVi over methods that segment objects using optical flow as input, is that SAVi can be applied to static images where optical flow is not available. We

demonstrate generalization to static images by evaluating 'SAVi + Bounding box' models on the first frame of all videos in the validation set. To this end, we repeat the first frame twice to form a "boring" video to test whether the model can accurately segment and "track" objects even in the absence of motion. In this setting, our model achieves a FG-ARI score of $88.7 \pm 0.2$ on the MOVi++ dataset, i.e. it accurately segments the static scene.

**Tracking subsets of objects** We investigate interfacing with SAVi to track a subset of the objects present in the first frame. We achieve this by conditioning slots using bounding boxes/segmentation masks for some of the objects and ignoring the bounding boxes/segmentation masks for the remaining objects. We found it beneficial, for these experiments, to initialize unconditioned slots with random vectors, so that they are free to model any remaining objects. We also found it beneficial to expose the model to this scenario at training time. We encoded whether a slot received conditional input or not using a binary indicator which we appended to the initial slot value.

Figure A.10 reports mIoU across various experimental conditions. The top two plots study SAVi under bounding box conditioning and the bottom two plots study SAVi under segmentation mask conditioning.

*Varying subset sizes for evaluation:* For the right two plots in Figure A.10, we train SAVi with up to 6 objects and evaluate with varying subset sizes. We observe that, selecting and tracking subsets of objects is indeed possible, but the introduction of randomly initialized slots comes at a price: with more objects without a conditioning signal, therefore needing to be explained by randomly initialized slots, the model's ability to connect the provided hints to the correct objects decays, as can be observed by a decrease in the mIoU score.

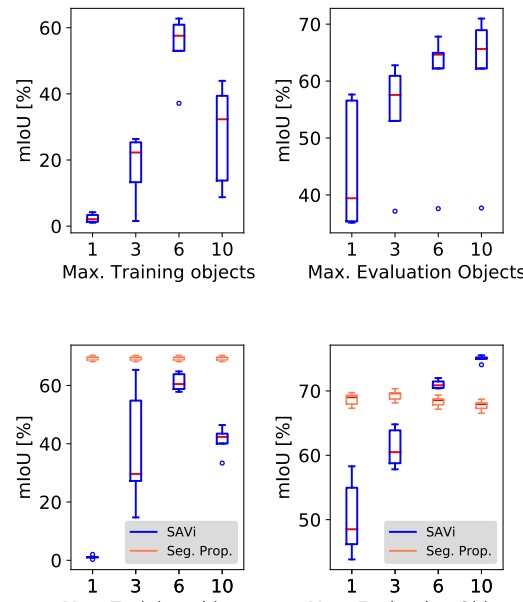

Figure A.10: Subset selection experiments: We select a subset of objects at train and test time and measure tracking mIoU on the MOVi dataset (first 6 frames only). Five seeds were used for each of these box-whisker plots.

*Varying subset sizes for training:* In the left two plots in Figure A.10, we train SAVi with various subsets of size $s \in \{1, 3, 6, 10\}$ and evaluate on 3 objects at test time. By training on subsets of objects, we are able to ablate between an unsupervised and a fully conditioned setting. As seen in the plots, when 'maximum training objects' is 1, that is 10 randomly initialized slots and 1 conditioned slot are used, the model is not able to establish a correspondence between the conditioning hint and the corresponding object in the scene. In other words, it treats the conditioning hint similar to the randomly initialized slots. On the other hand, providing conditioning for all objects at training time, that is 'maximum training objects' is 10, leads to poor generalization to subsets at test time since the model was not exposed to unconditioned objects during training. Conditioning on up to 6 objects presents a reasonable trade off between these two extremes. Providing explicit supervision on the alignment of the hint and the corresponding object in the input (which we do not provide) is likely necessary for stabilizing this setup when only few hints are provided. Alternatively, smarter initialization strategies for unconditioned slots can likely improve this setup, which we leave for future work.

**Generalization to unseen objects and backgrounds** To test for generalization capabilities, we evaluate a SAVi model with bounding box conditioning, which was trained on MOVi++, on the following evaluation splits (see Table A.1):

- **Unseen objects**: This split only contains novel objects (approx. 100) not seen during training, which are also not part of the default evaluation split. Remarkably, SAVi generalizes without a decrease in FG-ARI or mIoU scores on this split.

- **Unseen objects**: This split only contains novel backgrounds (approx. 40) not seen during training, but otherwise contains solely known objects. Generalization is comparable as to the default evaluation set.

- **Unseen objects & backgrounds**: When evaluating with both novel objects and backgrounds, there is only a slight decrease in FG-ARI and mIoU scores, i.e. the model still generalizes well.

- **MOVi**: A model trained on MOVi++ also generalizes well to MOVi, here at a resolution of $128 \times 128$ to be in line with the training setup on MOVi++.

Table A.1: Generalization results for a SAVi + Bounding Box model trained on MOVi++. Evaluation on first 6 video frames. Mean and standard error (10 seeds). All values in %.

| Evaluation split | FG-ARI↑ | mIoU↑ |
|---|---|---|
| MOVi++: Default | $82.0_{\pm 0.1}$ | $54.3_{\pm 0.3}$ |
| MOVi++: Unseen objects | $82.0_{\pm 0.1}$ | $54.4_{\pm 0.3}$ |
| MOVi++: Unseen backgrounds | $82.2_{\pm 0.1}$ | $54.0_{\pm 0.3}$ |
| MOVi++: Unseen objects & backgrounds | $80.4_{\pm 0.1}$ | $52.6_{\pm 0.3}$ |
| MOVi (trained on MOVi++) | $83.7_{\pm 0.2}$ | $53.7_{\pm 0.6}$ |

**Unconditional video decomposition on MOVi / MOVi++** In Table A.2 we report results on unconditional video decomposition on the MOVi and MOVi++ datasets. For SIMONe, we use our own reimplementation for which we approximately reproduced the CATER results reported by Kabra et al. (2021). For SCALOR (Jiang et al., 2020), we use the implementation provided by the authors. Further baseline details are provided in Section A.7.

Table A.2: Unconditional video decomposition on MOVi / MOVi++.

| | MOVi | MOVi++ |
|---|---|---|
| **Model** | **FG-ARI↑** | **FG-ARI↑** |
| SCALOR | $81.2_{\pm 0.2}$ | $22.7_{\pm 0.9}$ |
| SIMONe | $74.8_{\pm 4.2}$ | $32.7_{\pm 2.3}$ |
| SAVi (uncond.) | $78.2_{\pm 0.7}$ | $47.6_{\pm 0.2}$ |

**Architecture ablations** In this section, we report results on several architecture ablations on SAVi. In all cases, we evaluate on MOVi++ and condition SAVi on bounding box information in the first frame of the video. Results are reported in Table A.3. We consider the following ablations:

- **No predictor**: Instead of a Transformer-based (Vaswani et al., 2017) predictor, we use the identity function. Modeling the dynamics of the objects is thus entirely offloaded to the corrector, which does not model interactions. We find that this can marginally increase FG-ARI scores, likely because the self-attention mechanism in the default model variant can have a regularizing effect. It, however, negatively affects the model's ability to learn the correspondence between provided hints and objects in the video, as indicated by the drop in mIoU scores, likely due to its inability to exchange information between learned slot representations in the absence of the Transformer-based predictor.

- **MLP predictor**: In this setup, we replace the predictor with an MLP with residual connection and LayerNorm (Ba et al., 2016), applied independently on each slot with shared parameters. It has the same structure and number of parameters as the MLP in the Transformer block used in our default predictor. Similar to the *no predictor* ablation, this variant does not model interactions between slots. The effect on performance is comparable to the *no predictor* ablation.

- **Inverted corrector attention**: By default, we use Slot Attention (Locatello et al., 2020) as our corrector. The attention matrix in Slot Attention is softmax-normalized over the slots,

which creates competition between slots so that two slots are discouraged to attend to the same object or part (unless their representations are identical). In this ablation, we normalize the attention matrix instead over the visual input features, as done in e.g. RIM (Goyal et al., 2021c). We observe a sharp drop in the model's ability to learn the correspondence between hints and objects, as indicated by the low mIoU score in this setting.

Table A.3: SAVi ablations and model variants on MOVi++ with bounding box conditioning. Evaluation on first 6 video frames. Mean and standard error (10 seeds). All values in %.

| Model variant | FG-ARI↑ | mIoU↑ |
|---|---|---|
| SAVi (default) | $82.0 \pm 0.1$ | $54.3 \pm 0.3$ |
| No predictor | $83.0 \pm 0.2$ | $44.7 \pm 3.4$ |
| MLP predictor | $82.8 \pm 0.2$ | $49.5 \pm 2.2$ |
| Inverted corrector attention | $78.3 \pm 0.4$ | $11.4 \pm 0.4$ |

**Comparison to semi-supervised training with matching**   While our approach of conditioning the model on a set of hints, such as bounding boxes, in the first frame can be seen as a form of semi-supervised learning, SAVi is still solely trained with an unsupervised flow prediction (or reconstruction) objective. Instead of providing bounding boxes as conditional input to initialize the slots of SAVi, an interesting variant is to instead initialize slots in an unconditioned way and provide the bounding boxes instead as supervision signal in the form of a matching-based loss during training. This variant receives the same information as our conditional setup, but now requires matching to associate slots to bounding box labels. We train the model in the same way as the supervised set prediction set up of Slot Attention using Hungarian matching and a Huber loss as described by Locatello et al. (2020) with labels provided only during the first frame of the video. Additionally we still decode optical flow predictions at every time step using the same loss as in our default model. We found that this setup requires more than a single iteration of Slot Attention to converge to a sensible decomposition in the first frame where we apply matching. Results are summarized in Table A.4. We find that using matching as training signal results in significantly worse FG-ARI segmentation score at evaluation time compared to using conditioning both at training and at test time.

**Semantic property readout**   To evaluate to what degree slots of the SAVi model capture information about objects beyond their location and velocity, we train a simple readout probe in the form of an MLP that is tasked to predict object properties given a learned slot representation in SAVi. For these experiments, we train SAVi with bounding box conditioning on MOVi and jointly train an MLP readout head to predict object properties from each slot representation in the first frame. We do not propagate the gradients from the readout head through the rest of the model, i.e. the SAVi model does not receive any supervision information about object properties. The MLP readout head has a single hidden layer with 256 units and a ReLU activation. It is tasked to predict classification logits for (i) the object color out of 8 possible colors, (ii) the object shape out of 3 possible shapes, and (iii) the object material out of 2 possible materials. For each of the three classification targets we train with a cross-entropy loss using all videos of the MOVi training set. For evaluation, we measure average accuracy where each classification target (color, shape, material) is considered separately and the result is averaged over videos and classification targets.

In this setting, the readout probe achieves an accuracy of $54.6 \pm 1.5$ (10 seeds, mean and standard error) on the MOVi evaluation set which is far above chance level, indicating that semantic information

Table A.4: SAVi on MOVi with bounding box conditioning vs. unconditional SAVi with bounding boxes provided as supervision during the first fame via a matching-based loss. Evaluation on first 6 video frames. Mean and standard error (10 seeds). All values in %.

| Model variant | FG-ARI↑ |
|---|---|
| Conditional SAVi (default) | $92.9 \pm 0.1$ |
| SAVi + Learned init. + Matching loss | $83.0 \pm 0.2$ |
| SAVi + Gaussian init. + Matching loss | $82.3 \pm 0.7$ |

about objects is still to some degree retained in the slots even though our training objective (optical flow) is insensitive to color and material appearance. When training with RGB frame reconstruction instead, we can achieve a significantly higher score of $85.7 \pm 0.7$, suggesting that one could likely combine the benefits of frame reconstruction and flow prediction by training a model to jointly predict both targets, which is a promising direction for future work.

## A.5 DATASET DETAILS

The Kubric (Greff et al., 2021) dataset generation pipeline is publicly available under an Apache 2.0 license. MOVi++ contains approx. 380 publicly available CC-0 licensed HDR backgrounds from `https://hdrihaven.com/`. The data does not contain personally identifiable information or offensive content.

The original CATER (Girdhar & Ramanan, 2019) dataset (without segmentation mask annotations) is publicly available under an Apache 2.0 license. The variant with segmentation masks was provided by the authors of SIMONe (Kabra et al., 2021).

## A.6 ARCHITECTURE DETAILS AND HYPERPARAMETERS

All modules use shared parameters across time steps, except for the initializer, which is only applied at the first time step.

**Encoder** Our encoder architecture is inspired by the one used in Slot Attention (Locatello et al., 2020). We summarize the architecture in Table A.5. We use the same linear position embedding as in Slot Attention, i.e. a four-dimensional vector per spatial position encoding the coordinate along the four cardinal directions, normalized to $[-1, 1]$. We project it to the same size as the feature maps of the CNN encoder using a learnable linear transformation. The projected position embedding is then added to the feature maps. For the modified *SAVi (ResNet)* model on MOVi++, we replace the convolutional backbone (the first 4 convolutional layers) with a ResNet-34 (He et al., 2016) backbone. We use a modified ResNet root block without strides (i.e. $1 \times 1$ stride), resulting in $16 \times 16$ feature maps after the backbone. We further use group normalization (Wu & He, 2018) throughout the ResNet backbone.

Table A.5: SAVi encoder architecture for MOVi and CATER with $64 \times 64$ input resolution. For MOVi++ with $128 \times 128$ input resolution, we use a larger input stride of 2 and 64 channels (numbers in parentheses).

| Layer | Stride | #Channels | Activation |
|---|---|---|---|
| Conv $5 \times 5$ | $1 \times 1$ ($2 \times 2$) | 32 (64) | ReLU |
| Conv $5 \times 5$ | $1 \times 1$ | 32 (64) | ReLU |
| Conv $5 \times 5$ | $1 \times 1$ | 32 (64) | ReLU |
| Conv $5 \times 5$ | $1 \times 1$ | 32 (64) | – |
| Position Embedding | – | 32 (64) | – |
| Layer Norm | – | – | – |
| Conv $1 \times 1$ | $1 \times 1$ | 64 | ReLU |
| Conv $1 \times 1$ | $1 \times 1$ | 64 | – |

**Corrector** For the Slot Attention (Locatello et al., 2020) corrector, we only use a single iteration of the attention mechanism in all experiments on MOVi and MOVi++, and two iterations on CATER experiments. The query/key/value projection size is $D = 128$. Before each projection, we apply Layer Norm (Ba et al., 2016). The attention update is followed by a GRU (Cho et al., 2014) that operates on the slot feature size of 128 (unless otherwise mentioned). Locatello et al. (2020) further describe placing an optional feedforward layer after the GRU, which we only use when using more than a single iteration of Slot Attention: this MLP has a single hidden layer of size 256.

**Predictor** We use the default multi-head dot-product attention mechanism (MultiHeadSelfAttn) from Vaswani et al. (2017) as part of our predictor, for which we use a query/key/value projection

size of 128 and a total of 4 attention heads. For the MLP, we use a single hidden layer with 256 hidden units and ReLU activation. On CATER, we use the identity function as predictor, as objects do not interact in this dataset.

**Decoder**   We use the same decoder architecture as in Slot Attention (Locatello et al., 2020), for which we specify the details in Table A.6. Each slot representation is first spatially broadcasted to an $8 \times 8$ grid (i.e., the same vector is repeated at each spatial location) and then augmented with position embeddings which are computed in the same way as in the encoder. We use a small CNN to arrive at representations with the same resolution as the input frame. Finally, the output of this CNN is processed by two projection heads in the form of $1 \times 1$ convolutions, (1) to read out the segmentation mask logits and (2) to read out the per-slot predicted optical flow (or reconstructed RGB image). The segmentation mask logits are normalized using a softmax (across slots) and then used to recombine the per-slot predictions into a single image. The decoder uses shared parameters across slots and time steps. Note that the ground-truth optical flow is computed w.r.t. the previous frame, using an additional simulated frame before the first time step (which is only used for computation of ground-truth optical flow and later discarded).

Table A.6: SAVi decoder architecture for MOVi and CATER with $64 \times 64$ resolution. For MOVi++ with $128 \times 128$ resolution, we use a larger output stride of 2 (numbers in parentheses).

| Layer | Stride | #Channels | Activation |
|---|---|---|---|
| Spatial Broadcast $8 \times 8$ | – | 128 | – |
| Position Embedding | – | 128 | – |
| ConvTranspose $5 \times 5$ | $2 \times 2$ | 64 | ReLU |
| ConvTranspose $5 \times 5$ | $2 \times 2$ | 64 | ReLU |
| ConvTranspose $5 \times 5$ | $2 \times 2$ | 64 | ReLU |
| ConvTranspose $5 \times 5$ | $1 \times 1$ $(2 \times 2)$ | 64 | – |

**Initializer**   To encode bounding boxes and center of mass coordinates we pass the coordinate vector for each bounding box or center of mass location through an MLP with a single hidden layer of 256 hidden units, a ReLU activation and an output size of 128. The MLP is applied with shared parameters on each bounding box (or center of mass location) independently. Bounding boxes are represented as $[y_{\min}, x_{\min}, y_{\max}, x_{\max}]$ coordinates and center of mass locations are represented as $[y, x]$. Any unconditioned slot is provided with $[-1, -1, -, 1-1]$ bounding box coordinates ($[-1, -1]$ for center of mass conditioning).

When conditioning on segmentation masks, we use a CNN encoder applied independently on the binary segmentation mask for each conditioned foreground object. For each unconditioned slot, we provide an empty mask filled with zeros as input to the CNN. The architecture is summarized in Table A.7.

**Other hyperparameters**   As described in the main paper, we train for 100k steps with a batch size of 64 using the Adam optimizer (Kingma & Ba, 2015) with a learning rate of $2 \cdot 10^{-4}$ and gradient clipping with a maximum norm of 0.05. Like in previous work (Locatello et al., 2020), we use learning rate warmup and learning rate decay. We linearly warm up the learning rate for 2.5k steps and we use cosine annealing (Loshchilov & Hutter, 2017) to decay the learning rate to 0 throughout the course of training. Like prior work (Locatello et al., 2020), we use a total of 11 slots in SAVi for our scenes which contain a maximum number of 10 objects. In the conditional setup, all unconditioned slots share the same initialization and hence effectively behave as one single slot, irrespective of their number. Our chosen hyperparamters are close to the ones used in prior work and we did not perform an extensive search. We chose learning rate and gradient clipping norm based on performance measured on a separately generated instance of the MOVi++ dataset. We found that training with a substantially longer schedule (e.g. 1M steps) can still significantly improve results for SAVi, but we opted for shorter schedules to allow for easier and faster reproduction of our experiments. We only train the modified *SAVi (ResNet)* model on MOVi++ with a longer schedule of 1M steps, which we found to be especially beneficial for this significantly larger model.

Table A.7: SAVi initializer architecture for segmentation mask conditioning signals. We use the same architecture for all datasets. After the initial convolutional backbone, we perform a spatial average, followed by Layer Norm (Ba et al., 2016) and a small MLP, to arrive at the initial slot representations.

| Layer | Stride | #Channels | Activation |
|---|---|---|---|
| Conv $5 \times 5$ | $2 \times 2$ | 32 | ReLU |
| Conv $5 \times 5$ | $2 \times 2$ | 32 | ReLU |
| Conv $5 \times 5$ | $2 \times 2$ | 32 | ReLU |
| Conv $5 \times 5$ | $1 \times 1$ | 32 | – |
| Position Embedding | – | 32 | – |
| Layer Norm | – | – | – |
| Conv $1 \times 1$ | $1 \times 1$ | 64 | ReLU |
| Conv $1 \times 1$ | $1 \times 1$ | 64 | – |
| Spatial Average | – | – | – |
| Layer Norm | – | – | – |
| Dense | – | 256 | ReLU |
| Dense | – | 128 | – |

**Training software and hardware**   We implement both SAVi and the T-VOS baseline in JAX (Bradbury et al., 2018) using the Flax (Heek et al., 2020) neural network library. We train our models on TPU v3 hardware. We report training time on GPU hardware in Section 4.

## A.7   BASELINE DETAILS

**SCALOR**   This baseline (Jiang et al., 2020) patches the image into a $d \times d$ grid and uses a discovery network to propose objects in each grid cell. Then based on a threshold $\tau$ on the overlap of the proposed object with the propagated objects from the previous frames, the model rejects or accepts the proposals. The image is reconstructed with a combination of discovered objects and a background decoder with an encoding bottleneck dimension of $D_{\text{bg-en}}$. It is also important to have a correct mean and variance of the scale and ratio for the proposed objects. We performed a hyper-parameter search over the mentioned parameters along with the priors and annealing schedules for discovery and for the proposal rejection. We found the foreground-background splitting behavior to be especially sensitive to hyper-parameters, frequently resulting in the reconstruction of the entire image including all (or some) foreground objects with the background decoder, as opposed to the desired case where the background decoder only reconstructs the background of the scene. We also found that changing SCALOR's prior annealing schedule made the model highly unstable. We performed validation on held out $1/10$th of the training dataset, choosing the variants with best log likelihood and visual quality. All the hyper-parameters for the reported results are similar to the original SCALOR, except we use a $4 \times 4$ grid, a rejection threshold of 0.2, glimpse size of 32, scale mean and variance 0.2 and 0.1, object height-width ratio mean and variance 1.0 and 0.5, and for MOVi we decrease the background encoding dimension from the default 10 to 2. We use the official implementation provided by the authors, Jiang et al. (2020).

**SIMONe**   SIMONe (Kabra et al., 2021) is a probabilistic video model which encodes frames using CNNs followed by a transformer (Vaswani et al., 2017) encoder. It subsequently produces per-frame embeddings and per-object embeddings by pooling latent representations over (per-frame) transformer tokens and over time, respectively. Each pixel (for each time step and for each object embedding) is decoded independently using an MLP, conditioned on both time and object embedding (sampled from a learned posterior). The model is trained using a reconstruction loss in pixel space (in the form of a mixture model likelihood) and a regularization loss on the latent variables. SIMONe encodes all frames in a video in parallel, i.e. it differs from all other considered video baselines in that it is not auto-regressive and thus cannot generalize beyond the (fixed) clip length provided during training. We re-implement the SIMONe model in JAX (as there is no public implementation by the authors at the time of writing) with several simplifications: 1) we sample latent variables only once per latent variable and time step instead of re-sampling for every generated pixel, 2) we always decode all frames during training instead of sampling random sub-sets of frames, 3) we only train for 200k steps as we found that performance did not significantly improve with a longer schedule. We verified

that our re-implementation reaches comparable performance to the numbers reported by Kabra et al. (2021) on CATER. We otherwise utilize the same hyperparameters as reported by Kabra et al. (2021) for CATER in all our experiments, and we re-size all video frames to $64 \times 64$ before providing them to the model.

**Contrastive Random Walk (CRW)**    CRW (Jabri et al., 2020) is a self-supervised technique that induces a contrastive loss between embeddings of patches in neighboring frames. These embeddings are effective for downstream tracking via label propagation. We used the open source implementation provided by the authors to train their model on our datasets, MOVi and MOVi++; and evaluated tracking performance using the Davis 2017 benchmark utilities (Caelles et al., 2019; Pont-Tuset et al., 2017) on the validation sets of MOVi and MOVi++ respectively. We resized all images to $256 \times 256$ for training and to $480 \times 480$ for evaluation. This matches the setup that CRW was originally trained and tested on (Kinetics at $256 \times 256$ and Davis at 480p). Label propagation is more accurate at high resolution. We tuned and set the following parameters for MOVi++: Edge-dropout rate 0.05, training temperature 0.01, evaluation temperature 0.05, frame sampling gap 1, number of training epochs 250 with a learning rate drop at 200 epochs. And for MOVi: Edge-dropout rate 0.05, training temperature 0.03, evaluation temperature 0.01, frame sampling gap 2, number of training epochs 40 with a learning rate drop at 30 epochs. Early stopping was necessary for MOVi as training longer hurt downstream performance. We used video clips consisting of 6 frames to train these models. All other hyper-parameters and the architecture, a ResNet-18 (He et al., 2016) encoder, were kept at their default values.

**Transductive VOS**    Transductive VOS (T-VOS) (Zhang et al., 2020) propagates segmentation labels from the first frame into future frames, thus tracking objects and segmenting them. It correlates per-pixel features of a target frame with those of a history of previous frames, called reference. Softmax-normalized correlation weights are used to compute a convex combination of previous frame labels. This constitutes the prediction for each pixel in the target frame. Such predictions in turn get incorporated into the reference for subsequent label propagation. A simple motion prior is used to focus on neighboring regions when propagating labels. T-VOS uses references frames to capture the past video context. For T-VOS, we use the official implementation by the authors, using an ImageNet pre-trained ResNet-50 (He et al., 2016) backbone excluding the last stage for better results. We upsample MOVi/MOVi++ frames to 480p resolution, as we found that T-VOS with a ResNet-50 backbone produces inferior results when applied on frames at the original resolution. Metrics were computed at the original resolution of the two datasets. We select the optimal T-VOS temperature hyperparameter ($\tau = 1$ for MOVi and $\tau = 1.3$ for MOVi++) based on evaluation performance.

**Segmentation Propagation**    For a more direct comparison to SAVi, we apply T-VOS (Zhang et al., 2020) on a backbone that is pre-trained using the same data and optical flow supervision as SAVi. For this experiment, we use a simplified re-implementation of T-VOS in JAX, for which we use a maximum of 9 reference frames of the immediate past. We do not make use of the sparse sampling method from (Zhang et al., 2020). This is similar to the "9 frames" column in Table 2 as reported by Zhang et al. (2020), but in addition also uses their motion prior. We trained a self-supervised CNN backbone, in domain, using MOVi/MOVi++ frames. The self-supervision proxy task was that of single image optical flow prediction, inspired by prior work on predicting optical flow from static images (Mahendran et al., 2019; Walker et al., 2015). This proxy task requires guessing object motion based on appearance, which in turn requires modeling the objects present in the scene.

In more detail, we regressed optical flow using a fully convolutional network (Long et al., 2015) using the exact same loss function as SAVi. The image backbone was also identical to SAVi except for an extra $5 \times 5$ convolution layer to project features into 128 dimensions. This matches the dimensionality of slot vectors in SAVi making this a fairer comparison.

To pre-train the visual backbone, we use a batch size of 64 and we train for 100k steps using the Adam optimizer (Kingma & Ba, 2015) with a learning rate of $2 \times 10^{-5}$. We use linear position embeddings in the same way as SAVi, adding them right before a $1 \times 1$ convolution layer which regresses 3-dimensional RGB encoded optical flow. For evaluation of T-VOS, we use their motion prior with $\sigma_1 = 2.3, \sigma_2 = 4.6$, and a temperature of 50 for MOVi and 200 for MOVi++. We use $L2$-normalized per-pixel embeddings in order to make T-VOS less sensitive to temperature.

## A.8 METRIC DETAILS

For both FG-ARI and mIoU, we skip the first frame during evaluation as we provide conditional information (e.g., bounding boxes) for it in most cases.

**FG-ARI**   For our experiments on CATER (unsupervised video decomposition) we follow prior work (Kabra et al., 2021) and report FG-ARI for all frames.

**Segmentation mIoU**   This metric assumes a strict alignment between ground truth and predictions. That is, if the first segment tracks a ball then the first slot should track the same ball. No matching is used. Unlike ARI, this metric is sensitive to slots tracking the object they were conditioned with. In frames were an object is missing, either due to occlusion or because it has left the scene, $\mathrm{mIoU} = 1.0$ if the object is missing in the prediction as well, otherwise $\mathrm{mIoU} = 0.0$. Thus even a single predicted pixel in such frames is penalized heavily. This setting corresponds to the Jaccard-Mean metric used in the DAVIS-2017 object tracking challenge (Caelles et al., 2019; Pont-Tuset et al., 2017).

