# OpenReview forum: "Conditional Object-Centric Learning from Video"
_ICLR.cc/2022/Conference — ICLR 2022 Poster_

### Official Review · Reviewer_TdU6 · 2021-10-21

**Correctness:** 4
**Technical Novelty And Significance:** 4
**Empirical Novelty And Significance:** 4
**Recommendation:** 8
**Confidence:** 5

**Main Review:**

I like this paper overall, and believe that the two improvements would bring new knowledge to the object-centric learning community.

Strengths
1. Previous object-centric learning methods are mostly unsupervised. The visual results they generate on simple datasets like CATER are quite promising, but in fact, they often fail in outputting meaningful object segmentations. For instance, an object can be represented by two slots while we only want a single one. I think this can hardly be well-addressed under the pure unsupervised setting. To this end, this work shows giving a simple initialization of objects of interest leading to less ambiguous representations of object slots, and I think this is a promising direction for object-centric learning.
2. Using optical flows as supervision makes it possible to perform object-centric learning on more realistic datasets. In such a dataset, the texture and color of objects are more diverse and the previous RGB reconstruction becomes an inappropriate pre-text task: We do not need to remember every detail of the objects' appearance in order to track them. Instead, predicting optical flows is an easier and more beneficial pre-text task.
3. I really like the experimental setups and results. The extensive experiments successfully convince me about the effectiveness of the two improvements proposed.

Weakness:
1. I think the biggest limitation is that using optical flow as supervision cannot handle static objects. However, this is clearly discussed in the limitation section, and I agree that addressing this problem is out of the scope of this paper.

Minor concerns:
1. Section 2,  "Inspired by ... ordinary differential equations", missing references.
2. In the legend of Figure 1, the text "Gated Recurrent Units" is not aligned.
3. Section 2, the Encoder paragraph, the notation is not clear. What is N and D_{enc}?
4. Section 2, the Slot Initialization paragraph, "we set the conditional input to a fixed value indicating padding": what does "indicating padding" mean here?




**Summary Of The Paper:**

This paper considers the problem of learning object-centric representations from videos. Different from existing methods that mostly model this problem under a pure unsupervised setting by reconstructing video frames, this work introduces two improvements. The first is offering weak "hints" (can be seen as additional inputs) during training, which can avoid falling into a trivial solution like previous unsupervised approaches always did. The "hints" here can be in form of pixel-wise masks, bounding boxes, or even as simple as centers of mass.  The second improvement regards the supervision signals. Instead of reconstructing raw pixels, they propose to predict optical flows. In easy scenes the optical flows are actually similar to pixel-wise masks: values inside an object are prone to be consistent. Therefore the learning is eased, and then making it possible to train on more realistic datasets. Extensive experiments are conducted to validate the effectiveness of these two improvements.

**Summary Of The Review:**

This paper introduces two improvements upon existing object-centric learning methods: offering additional object-level "hints" as inputs and using motion cues (optical flows) as pre-text supervision. The idea is good and I believe will be beneficial to the object-centric learning community. The writing is good, easy to follow. Experimental results are sufficient to support their claims.

---

> ### Author Response · Authors · 2021-11-12
> **Response to Reviewer TdU6**
>
> Thank you for your review and for your encouraging comments.
>
> We are very happy to see your positive assessment of our paper. Thank you for explicitly highlighting your view on the strengths of our paper in such great detail.
>
> **Q1: “Section 2, the Encoder paragraph, the notation is not clear. What is N and D_{enc}?”**
>
> Thank you for pointing this out, we will clarify this in the paper. N is the size of the flattened grid, i.e. width*height of the encoder feature maps, whereas D_{enc} is the feature dimensionality (per grid location) of the representations returned by the encoder, i.e. its feature map dimensionality.
>
> **Q2: “I think the biggest limitation is that using optical flow as supervision cannot handle static objects.”**
>
> This is indeed the case at training time, and (as you mention) one of the limitations we discuss in the paper. Addressing this limitation is an important direction for future work. Note however, that SAVi trained with optical flow can nonetheless generalize to static scenes / static objects at test time, as demonstrated in appendix section A.2.
>
>
> **Q3: “Section 2, the Slot Initialization paragraph, "we set the conditional input to a fixed value indicating padding": what does "indicating padding" mean here?”**
>
> Thanks, we will clarify this in the paper. We always use the same number of slots in SAVi for all videos in the dataset, irrespective of the number of objects in the particular video. In the conditional setting, the number of conditional cues K (e.g. bounding boxes) can vary between videos and we pad this conditional input always to the same shape, i.e. the first K slots will receive a “true” conditional input, whereas the remaining slots will receive a fixed value which indicates that their input is “padding”. We do this so that we can efficiently construct mini-batches, but other more complicated schemes, e.g. using explicit masking, would be possible as well.
>
>
> Thank you also for your feedback regarding figure legend alignment and the missing references on predictor-corrector methods for ODEs. Surprisingly, there is no canonical reference for predictor-corrector methods (or at least, we could not find one). This is likely because predictor-corrector methods are frequently considered common background knowledge in numerical analysis. We wanted to avoid simply pointing to a Wikipedia article or a (typically not freely accessible) book on the subject of ODEs (e.g. [1]), but we are happy to reconsider this decision.
>
> [1] Butcher, “Numerical Methods for Ordinary Differential Equations” (2003)

---

### Official Review · Reviewer_r9zz · 2021-10-29

**Correctness:** 4
**Technical Novelty And Significance:** 3
**Empirical Novelty And Significance:** 3
**Recommendation:** 6
**Confidence:** 5

**Main Review:**

Strengths:
1) Overall, this paper is well written, and the technical details are easy to follow.
2) The main idea of learning object representations and physical dynamics from videos is interesting.
3) I found the anecdotal evidence for segmenting and tracking corresponding parts of objects very interesting, opening the door for more hierarchical concepts of objects using self/semi-supervised approaches.

Weaknesses:

**Contributions.** [1] already proposed a Slot Attention model based on optical flow for segmenting a single object. Although the author’s method supports multi-object environments while [1] is not, this paper still did not evaluate the proposed approach on real-world data (unlike [1]), which is a concern to me.

Few works have already shown that using learnable query vectors instead of Gaussian-initialized slots helps the slots to learn on a unique embedding. [1] shows that “learnable queries play a similar role as soft clustering, i.e., assigning each pixel to one of the motion groups.” [2] (and others) shows that 2D positional encoding and box query embeddings are essential for initializing the queries for other tasks. To summarize, these works share the same concept that initializing the slots with location or motion embeddings could play a significant role in different downstream tasks.

[1] Self-supervised video object segmentation by motion grouping, ICCV 2021.

[2] TubeR: Tube-Transformer for Action Detection.


**Real-world Video Data.** As far as I see it, multi-object segmentation and tracking are essential for real-life datasets, and I am concerned that this approach could be relevant only for synthetic datasets. The main issue with the Slot attention model is the inability to capture natural texture/background and/or camera movement, and thus I believe this work will not be able to generalize in the real world.


**Technical Novelty.** The proposed approach is heavily based on the Slot Attention [2] model. I could not find an apparent and exciting technical novelty that could be interesting for other domains or different tasks. I believe the extension for the video domain could reach the bar of the conference quality, but still, I expect the authors to bring something new to the table.

Can the authors highlight 2-3 points of technical or architectural modeling that are unique and different from their perspective from [1] or [2]?

[1] Self-supervised video object segmentation by motion grouping, ICCV 2021.

[2] Object-centric learning with slot attention, NeurIPS 2020.


**Relation to Prior Work.** There are some object-centric approaches that use object-centric representations for video understanding and might be worth considering citing them, such as:

[*] Compositional Video Synthesis with Action Graphs, ICML 2021.

[**] Spatio-Temporal Action Graph Network, ICCVW2019.







**Summary Of The Paper:**

This paper introduces a sequential extension of Slot Attention to tackle the problem of unsupervised / weakly supervised multi-object segmentation and tracking in video data. The method demonstrates successful segmentation and tracking for synthetic video data on unsupervised object representation learning.


**Summary Of The Review:**

My main concern is that I cannot see how the proposed approach can generalize to new domains and tasks. Overall, I like this paper and its contribution, but I think the authors should clarify their contributions and explain how the video slot attention could be leveraged in other domains and tasks. Otherwise, it feels like a perfect model for the CATER/ CLEVRER/ MOVi/ MOVi++ datasets, but the full potential is not entirely clear.

I am open to the authors' feedback and other reviewers' opinions.


After Rebuttal
------------------


After reading the authors' feedback and other reviewers' opinions, I would like to thank the authors for their rebuttal.

The rebuttal addresses most of my concerns. I am leaning towards acceptance of the paper since it maintains the high bar of the conference quality. I vote for 6.

---

> ### Author Response · Authors · 2021-11-12
> **Response to Reviewer r9zz**
>
> Thank you for the detailed review and for your insightful comments.
>
> **Regarding our contributions:**
>
> It is indeed correct that Yang et al. [1] demonstrated an application of Slot Attention [2] for segmenting optical flow images into foreground vs. background. Our approach and the problem we consider, however, differ from theirs in a number of ways: while [1] only segments single video frames independently, we address the significantly more difficult problem of temporally-consistent segmentation of an entire video. For [1], the temporal consistency problem does not arise as they only consider foreground-background segmentation, which can be solved independently per video frame using an image segmentation model like Slot Attention and by limiting it to two latent slots with a fixed, learnable initialization. In the multi-object setting this is generally not possible, and requires some form of alignment between frames. We solve this problem by directly training on videos and by carrying latent slot information forward in time (combined with a transition model / “predictor”), which is a non-trivial extension of the Slot Attention framework. Another core difference between our approach and [1] is that we operate on textured, visual (RGB pixel) information, whereas the approach in [1] solely auto-encodes optical flow images and is, for example, not applicable to static scenes at test time (for which the optical flow input carries no information about objects). [1] would further not be able to learn about texture- or color-dependent properties of objects, as it solely relies on optical flow input.
>
> The setting (but not necessarily the data) we consider is significantly more difficult than the one considered in [1]. We believe that it is necessary to make progress in this more general multi-object setting (with textured, visual observations) to ultimately arrive at methods which can develop an object-centric understanding of diverse real-world video data. Our approach takes a significant step in this direction, but as we outlined in our Limitations section, there are still several obstacles that have to be overcome in future work for applicability to the full visual and dynamic complexity of the real world.
>
> Thank you for also highlighting your concern regarding learnable query vectors not being an original contribution, as recent works such as [1] and [3] similarly used learnable query vectors. We would like to emphasize that learnable query vectors are *not a claimed novel contribution of our work*. Slot Attention [2], which forms a core component of our model, is indeed frequently used with learnable slot initialization vectors (i.e. query embeddings). This was in fact already investigated in the original Slot Attention paper by Locatello et al. [2] (see appendix B “slot initialization”). One limitation of learnable query embeddings pointed out by Locatello et al. [2] is that “adding additional slots at test time is not possible without re-training” [2, appendix B]. Our *conditional* slot initialization scheme avoids this problem by a) learning a conditional initialization function that produces initial slot vectors given some external cue, such as a bounding box, and b) re-using this function across all slots. In other words, the embedding produced by our conditional initialization is not unique to the slot index (unlike default learnable query embeddings) and thus the number of conditioned slots can be flexibly changed both at training and at test time, which we demonstrate by conditioning on a varying number of object cues. Conditional initialization also serves as a convenient interface with the model, as it allows the user to specify which objects or parts of objects should be represented, tracked and segmented by providing small hints such as approximate center-of-mass coordinates or bounding box coordinates in the initial video frame.
>
> **Regarding real-world video data:**
>
> To alleviate your concern about real-world applicability of our approach, we added a fully-unsupervised qualitative experiment (i.e. without conditioning or optical flow information) on a real-world robotics dataset. We find that SAVi is able to decompose the scene into meaningful object components, while maintaining temporal consistency of the decomposition for unseen evaluation videos of up to 200 frames, something which is unachievable by prior unsupervised methods such as SIMONe, which we qualitatively compare against. You can find these qualitative video results in the supplementary zip file which we have added to our submission. As mentioned above, however, generalization to the full visual and dynamic complexity of the real world is still an open problem, but this experiment demonstrates that this class of models is not limited to simulated data only.

---

> > ### Author Response · Authors · 2021-11-12
> > **Response to Reviewer r9zz (Part 2)**
> >
> > **Technical Novelty: “Can the authors highlight 2-3 points of technical or architectural modeling that are unique and different from their perspective from [1] or [2]?”**
> >
> > As described above, we develop an extension of Slot Attention to video data that can be directly trained on videos by carrying latent slot information forward in time (combined with a transition model / “predictor”). This is different from [1] and [2], which are limited to training on single images / frames independently.
> > Conditioning: we demonstrate that providing small hints to the latent slots (as abstract as the 2D coordinates of a point on an object in the first frame) in the form of a novel conditional slot initialization module is sufficient to substantially improve object-centric learning, as the notion of an object can be ill-defined without this specification or context.
> > Lastly, we demonstrate that using optical flow as a self-supervision target for multi-object segmentation from raw visual, textured RGB inputs can overcome limitations of prior approaches in cases where objects are highly textured and moving at training time. No flow is needed at test time, unlike for [1], which allows our model to decompose textured static scenes at test time as well (see appendix A.2).
> >
> > Thank you also for your pointers to additional related work. We will reference these in our revision.
> >
> > [1] Yang et al., “Self-supervised Video Object Segmentation by Motion Grouping” (ICCV 2021)
> >
> > [2] Locatello et al., “Object-centric learning with slot attention” (NeurIPS 2020)
> >
> > [3] Zhao et al., “TubeR: Tube-Transformer for Action Detection” (2021)

---

> > > ### Comment · Reviewer_r9zz · 2021-11-20
> > > **Response to the Authors**
> > >
> > > After reading the authors' feedback and other reviewers' opinions, I would like to thank the authors for their rebuttal.
> > >
> > > The rebuttal addresses most of my concerns. I am leaning towards acceptance of the paper since it maintains the high bar of the conference quality. I vote for 6.

---

> > > > ### Author Response · Authors · 2021-11-22
> > > > **Re: Response to the Authors**
> > > >
> > > > Thank you for updating your review. We are happy to hear that we were able to address most of your concerns.

---

### Official Review · Reviewer_1UV9 · 2021-11-02

**Correctness:** 4
**Technical Novelty And Significance:** 3
**Empirical Novelty And Significance:** 3
**Recommendation:** 8
**Confidence:** 4

**Main Review:**

**Pros:**

- Adapting Slot Attention to produce coherent object-centric representations in video instead of static images is interesting and can lead to further development in the video-processing field. Moreover the idea is simple and sound.
- I enjoy the experiments showing the transition of results/empirical findings from simple to more complicated scenarios and also from synthetic to more realistic ones. The detailed ablations (built incrementally) offer useful insight for what the current object-centric video models require in order to perform well on more challenging scenarios.
- The experiments where the method is conditioned on a subset of objects (Appendix) set by the user are really nice and open a broad range of applications for the proposed method
- Overall the paper is well written and easy to follow

**Cons:**

- When measuring FG-ARI for the frame-level methods (Slot Att and MONet in Table1) how is the matching between the consecutives frames computed? I agree with the authors that the low score could be due to the lack of temporal consistency, but I think that a basic matching algorithm such as Hungarian matching + pairwise similarity to reorder the slots in a more appropriate way could represent a better baseline.
- It would be nice to see the results mentioned in the last paragraph of the Section 4.2 (Unsupervised settings on MOVi and MOVi++) in a table.
- In the paper, it is mentioned that the comparison with original CRW or T-VOS is not entirely fair since they used different backbone and operate on higher-resolution frames. I agree with that. However, why did the authors decide to retrain those methods instead of using the better backbone/setup for their method? I agree that the results should improve in that situation, but it’s not necessarily a sure fact.

**Summary Of The Paper:**

The paper proposes an object-centric method for video data, inspired from the slot-attention architecture. To enhance the learning (and to allow the users to specify the kind of entities that they are interested in), the slots are initialized from the first frame either a) random; b) with learnable initialisation; c) as the center of the bounding boxes; d) as bounding boxes or e) as segmentation maps.  The method uses a slot-attention mechanism to correct the slots from each frame, with query consisting in the current prediction of the slots, and keys and values consisting in some features extracted from the frame + GRU on each node, for temporal aggregation.  Then they apply a self-attention layer to propagate the information between slots. Finally, the model is optimise to predict either the optical flow or the rgb content.

The paper presents several ablations: using different types of conditioning, different levels of supervision (with or without optical flow or using estimated unsup optical flow instead of the real one); or by varying the amount of noise they inject in the initial positions of the slots (in the conditioned case).

The method is tested on CATER, MOVi and MOVi++ to quantify the quality of video decomposition and they also experiment with different types of generalization scenarios: various number of frames, new objects, new background, etc.

**Summary Of The Review:**

Overall, I enjoy the idea presented in the paper and I consider the empirical part interesting enough to recommend the acceptance.

---

> ### Author Response · Authors · 2021-11-12
> **Response to Reviewer 1UV9**
>
> Thank you for your review and for your very encouraging comments.
>
> We are happy to see your positive assessment of our paper and appreciate the detailed feedback.
>
> **Q1: “When measuring FG-ARI for the frame-level methods (Slot Att and MONet in Table1) how is the matching between the consecutives frames computed?"**
>
> Thank you for this question. Please note that the baseline results in Table 1a are obtained from Kabra et al. (NeurIPS 2021) [1] – as mentioned in the main text in Section 4.1. The values for Slot Attention and MONet represent performance when independently applying these methods per video frame, i.e. without a separate matching procedure between frames. While we agree that performance of these baselines can be improved by adding some form of temporal matching or a module that ensures temporal consistency, these baselines are intended to show what a direct application of the respective model to this dataset would yield in terms of segmentation performance. We very much appreciate the suggestion, however, and we will explain this detail in the paper. We believe that S-IODINE and SIMONe are good representatives of models in this domain that do account for temporal alignment and that this experimental comparison demonstrates the soundness of our architecture for the task of (fully) unsupervised object representation learning.
>
> **Q2: “It would be nice to see the results mentioned in the last paragraph of the Section 4.2 (Unsupervised settings on MOVi and MOVi++) in a table.”**
>
> This is a great suggestion, thank you. We will add a table with these results in the appendix.
>
>
> **Q3: “In the paper, it is mentioned that the comparison with original CRW or T-VOS is not entirely fair since they used different backbone and operate on higher-resolution frames. I agree with that. However, why did the authors decide to retrain those methods instead of using the better backbone/setup for their method?”**
>
> Thank you for this question! Our goal was to favor simplicity over complex (and compute-intensive) visual backbone engineering and to re-use the same backbone architecture across all experiments to ensure a fair comparison against earlier unsupervised methods on CATER. Hence this is only a conservative demonstration of the capabilities of the model.
>
> We fully agree, however, that investigating better backbone architectures for SAVi (in line with the CRW and T-VOS baselines) would add value to our experimental analysis. We ran an additional experiment on MOVi++ (our visually most complex dataset) where we replaced our simple CNN encoder with a more capable ResNet-34 encoder (using GroupNorm instead of BatchNorm, and without using strided convolution or max pooling in the stem to retain higher-resolution outputs of 16x16 feature maps). We trained this model using a longer schedule of 500k training steps (with otherwise identical hyperparameters) and we found that it performs on par with our best baseline (CRW) despite using only bounding boxes as conditioning signals (as opposed to segmentation masks for CRW). It achieves 50.7 ± 0.2 mIoU and 82.8 ± 0.4 FG-ARI. We will add this experiment to Table 1b.
>
> [1] Kabra et al., “SIMONe: View-Invariant, Temporally-Abstracted Object Representations via Unsupervised Video Decomposition” (NeurIPS 2021)

---

> > ### Comment · Reviewer_1UV9 · 2021-11-22
> > **Response to authors rebuttal**
> >
> > I would like to thank the authors for their rebuttal. After reading the rebuttal and in line with other reviewers’ opinion, I maintain my score, recommending the acceptance of the paper.

---

### Official Review · Reviewer_o4c6 · 2021-11-08

**Correctness:** 3
**Technical Novelty And Significance:** 3
**Empirical Novelty And Significance:** 3
**Recommendation:** 6
**Confidence:** 3

**Main Review:**

1. The weakly-supervised setting seems to be new and has not been used in the previous literacture, which adopts abstract hints of the first frame as input and achieves better performance.

2. The idea to use optical flow as supervision signal seems to be new and has not been done before.

3. The writing overall is easy to follow.

**Summary Of The Paper:**

This paper learns the object-centric representation for videos by extending the previous static slot attention framework with two new considerations, 1. optical flow for temporal modeling and 2. using simple objects' location cues for better segmentation or tracking. Experiments are conducted on CATER, MOVi and MOVi++. It show SAVi has better performance than its baselines especially when using the objects' cues during training.

**Summary Of The Review:**

The reviewer has some concerns on the novelty of the paper.

1. The idea of using optical flow to help weakly-supervised or unsuperised object learning seems to be straight forward. Previous work like PSGNet[A] also adopts motions and depth as supervision signal and compare their performance with MoNet. It will be necessary to compare and disucss the PSGNet in the related work.

2. The paper only compares with  Slot attention or MoNet in unsupervised object segmentation. Object-centric representation for video learning and reasoning has been widely studied in previous frameworks like [A, B, C, D]. Note that previous work like ALOE [B] and VRDP [C] has also been using Slot attention or MoNet for unsupervised video proposal segmentation. It will be interesting to replace the Slot attention or MoNet with the proposed segmentation model and see their performance on the CLEVRER[E] dataset.

[A]. Bear D M, Fan C, Mrowca D, et al. Learning physical graph representations from visual scenes[J]. arXiv preprint arXiv:2006.12373, 2020.

[B]. Attention over learned object embeddings enables complex visual reasoning. David Ding, Felix Hill, Adam Santoro, Malcolm Reynolds, Matt Botvinick, Arxiv, 2021.

[C]. Chen Z, Mao J, Wu J, et al. Grounding physical concepts of objects and events through dynamic visual reasoning[J]. arXiv 2021.

[D]. Ding M, Chen Z, Du T, et al. Dynamic Visual Reasoning by Learning Differentiable Physics Models from Video and Language[J]. arXiv 2021.

[E]. A]. Yi K, Gan C, Li Y, et al. Clevrer: Collision events for video representation and reasoning[J], ICLR 2020.

---

> ### Author Response · Authors · 2021-11-12
> **Response to Reviewer o4c6**
>
> Thank you for your review and for the extensive pointers to related works.
>
> We very much appreciate the suggestion of discussing PSGNet [1] in our paper. This is indeed a great paper and closely related work. We will reference PSGNet in our revision. We also appreciate the suggestion of explicitly comparing against their work. We would like to note, however, that PSGNet—as it is presented in the paper—is designed and evaluated as a method for image decomposition, but not for video decomposition. While the authors of PSGNet use temporal information from video clips (of 4 frames) during training, the method is only applied and tested on static images (unlike the experiments in our paper, which are focused on video evaluation). As we already include a comparison to two representative image decomposition models (Slot Attention and MONet), we don’t think that adding a comparison to a third such baseline would add much value for the readers of our paper, especially because our focus is on temporally consistent video decomposition and not (motion-informed) decomposition of individual images.
>
> Thank you also for the other suggestions regarding related work.
> * Please note that the paper by Ding et al. [2] appeared on arXiv *1 month after the ICLR 2021 submission deadline*. We are happy to cite this paper as concurrent work, but please understand that it would have been impossible for us to discuss this in our initial submission. The paper by Ding et al. [2] further does not address an un-/weakly-supervised setup, but instead uses a supervised Faster R-CNN model for detecting objects.
> * Ding et al. [3] use MONet for object segmentation/decomposition, which is one of the baselines we already compare against. Their supervised visual reasoning task on top of fixed MONet embeddings is otherwise very different from the tasks/settings we consider in our work.
> * Lastly, Chen et al. [4] use pre-trained (supervised) region proposal networks to extract objects on CLEVRER, i.e. they do not consider the problem of un-/weakly-supervised object-centric representation learning. We believe our comparison to state-of-the-art baselines (which also include S-IODINE and SIMONe [5]) on CATER is sufficient to establish the soundness and competitiveness of our fully unsupervised SAVi model variant. CLEVRER is visually very similar to CATER and – in the published version – does not come with segmentation masks, which makes evaluation difficult.
>
> We will add appropriate references to supervised visual reasoning work to our related work section – thank you again for these suggestions.
>
> Regarding your concern around our contribution of “using optical flow to help weakly-supervised or unsupervised object learning” being “straight forward”: we would like to highlight that no other work in un-/weakly-supervised multi-object learning has successfully used optical flow for learning temporally consistent multi-object decompositions in video using an end-to-end approach like ours. We believe that ideas like these can easily appear straightforward in hindsight, but experimentally demonstrating a successful implementation of this idea is difficult. The closest work in this regard is [6] which implements a strong simplification of this idea: it only performs foreground-background segmentation of single video frames (the application of this to multi-frame video at test time does not require object correspondence), does not decompose into multiple objects, and cannot be applied on RGB (non-flow) inputs, but instead requires optical flow as sole input.
>
>
> [1] Bear et al., “Learning physical graph representations from visual scenes” (NeurIPS 2020)
>
> [2] Ding et al., “Dynamic Visual Reasoning by Learning Differentiable Physics Models from Video and Language” (NeurIPS 2021)
>
> [3] Ding et al., “Attention over learned object embeddings enables complex visual reasoning” (NeurIPS 2021)
>
> [4] Chen et al., “Grounding physical concepts of objects and events through dynamic visual reasoning” (ICLR 2021)
>
> [5] Kabra et al., “SIMONe: View-Invariant, Temporally-Abstracted Object Representations via Unsupervised Video Decomposition” (NeurIPS 2021)
>
> [6] Yang et al., “Self-supervised Video Object Segmentation by Motion Grouping” (ICCV 2021)

---

> > ### Comment · Reviewer_o4c6 · 2021-11-20
> > **Review Updated.**
> >
> > The authors' rebuttal has resolved most of the reviewer's concerns. The reviewer is not an expert in video decomposition, which makes it a little bit hard for the reviewer to accurately estimate the paper's novelty.  After reading other reviewers' comments, the reviewer agrees to raise the rating to 6. Again, the reviewer still thinks this is a borderline paper.

---

> > > ### Author Response · Authors · 2021-11-22
> > > **Re: Review Updated.**
> > >
> > > Thank you for updating your review. We are happy to hear that we were able to resolve most of your concerns.

---

### Author Response · Authors · 2021-11-12
**General response**

We would like to thank the reviewers for their helpful feedback and insightful comments.

We are happy to see that the reviewers believe that our paper is *“well written”* (r9zz, 1UV9) and *“easy to follow”* (o4c6, 1UV9, r9zz). Our contribution of extending Slot Attention to video data to produce coherent temporal slot representations is described as *“interesting”* and as *”simple and sound”* (1UV9). Reviewer r9zz remarks that our empirical finding for steerable part-whole segmentation via conditioning is *“very interesting, opening the door for more hierarchical concepts of objects using self/semi-supervised approaches”*.

Reviewer o4c6 highlights the novelty of our weakly-supervised setting and the use of optical flow as a (self-)supervision signal for this class of models. With respect to our contribution of conditional / weakly-supervised learning, reviewer TdU6 writes: *“I think this is a promising direction for object-centric learning”*. Finally, reviewer TdU6 highlights that they *“really like the experimental setups and results”* and reviewer 1UV9 writes that our experiments on conditioning on a subset of object cues *“are really nice and open a broad range of applications for the proposed method”*, and that our ablation studies *“offer useful insight for what the current object-centric video models require in order to perform well on more challenging scenarios”*.

We further thank reviewers for the very thorough constructive feedback. In response to this feedback, we have conducted two new experiments which we summarize below:

**1. Application of SAVi to real-world video data (suggested by reviewer r9zz)**

To address the question whether SAVi is only applicable to synthetically generated simulated data or can be used in a real-world context, we trained the fully-unsupervised SAVi model variant on the real-world Sketchy [1] robotic grasping dataset using RGB reconstruction loss. We train SAVi on clips of 6 frames (i.e. the same as in our other experiments) and qualitatively evaluate the model on unseen test videos with a length of 200 frames. Our results show that SAVi can a) decompose these scenes into meaningful object components, and b) consistently represent and track individual scene components over long time horizons, far beyond what is observed during training. These video results can be found in our *newly uploaded supplementary zip file*, which we will refer to in our revision.

**2. Combining SAVi with stronger visual backbones (suggested by reviewer 1UV9)**

To investigate to what degree SAVi can benefit from stronger visual backbone architectures, we ran an additional experiment on MOVi++ (our visually most complex dataset) where we replaced our simple CNN encoder with a more capable ResNet-34 encoder. We trained this model using a longer schedule of 500k training steps (with otherwise identical hyperparameters) and we found that it performs on par with our best baseline (CRW) despite using only bounding boxes as conditioning signals (as opposed to segmentation masks for CRW). It achieves 50.7 ± 0.2 mIoU and 82.8 ± 0.4 FG-ARI, i.e. a significant improvement over the SAVi model with a simpler CNN backbone. We will add this experiment to Table 1b.

[1] Cabi et al., “Scaling data-driven robotics with reward sketching and batch reinforcement learning” (Robotics: Science and Systems Conference 2020)

We will upload a revised version of our paper incorporating the feedback by the reviewers and the experimental results mentioned above shortly. For other questions raised by the reviewers, please see our response to individual questions and concerns below each review.

---

> ### Author Response · Authors · 2021-11-18
> **Paper revision uploaded**
>
> We have uploaded a revised submission which includes two additional experiments (as discussed in our general response), various clarifications, and updates to our related work section to address the feedback by the reviewers.
>
> We'd like to thank the reviewers again for their constructive feedback.

---

### Decision · Program_Chairs · 2022-01-20

**Decision:**

Accept (Poster)

**Comment:**

This work proposes a new framework that can learn the object-centric representation for video. The authors did a good job during rebuttal and turned one slightly negative reviewer into positive ones. The final scores are 6,6,8,8. AC agrees that this work is very interesting and deserves to be published on ICLR. The reviewers did raise some valuable concerns that should be addressed in the final camera-ready version of the paper. The authors are also encouraged to make other necessary changes.